# Estimating the Probability of Compound Floods in Estuarine Regions

**Wenyan Wu [1,2], Seth Westra[2] and Michael Leonard[2]**

[1] The Department of Infrastructure Engineering, The University of Melbourne, 3010, Australia

[2] The School of Civil, Environmental and Mining Engineering, The University of Adelaide, 5005, Australia

*Correspondence to: Wenyan Wu (wenyan.wu@unimelb.edu.au)*

## Abstract

The quantification of flood risk in estuarine regions relies on accurate estimation of flood probability, which is often challenging due to the rareness of hazardous flood events and their multi-causal (or 'compound') nature. Failure to consider the compounding nature of estuarine floods can lead to significant underestimation of flood risk in these regions. This study provides a comparative review of alternative approaches for estuarine flood estimation; namely, traditional univariate flood frequency analysis applied to both observed historical data and simulated data, and multivariate frequency analysis applied to 'flood events'. Three specific implementations of the above approaches are evaluated on a case study — the estuarine portion of Swan River in Western Australia — highlighting the advantages and disadvantages of each approach. The theoretical understanding of the three approaches, combined with findings from the case study, enable generation of guidance on method selection for estuarine flood probability estimation, recognising issues such as data availability, complexity of the application/analysis process, location of interest within the estuarine region, computational demands and whether or not future conditions need to be assessed.

**Keywords:** Compound flood; Estuarine flood; Flood probability estimation

## 1 Introduction

Estimates of the probability of future floods represent a critical information source for applications such as land use zoning and planning, reservoir operation, flood protection infrastructure design and dam safety assessments (e.g. Ball et al. (2019)). Such probability estimates form the basis for calculations of the 'design flood' (a hypothetical flood with a defined probability of exceedance, such as the 1% annual exceedance probability flood or 1 in 100 years flood), as well as for risk-based approaches that consider the integration of both probability and consequence. Indeed, the estimation of flood probability represents one of the core objectives of the field of engineering hydrology (Maidment, 1993), with methodological developments dating back to early flood frequency estimation approaches (Condie and Lee, 1982; Riggs, 1966; Singh, 1980; Woo, 1971) and the development of rainfall intensity-frequency-duration (IFD) curves (Koutsoyiannis et al., 1998; Niemczynowicz, 1982; Yu and Chen, 1996).

Although many aspects of the flood probability calculation are strongly supported by theory and embedded in engineering practice (e.g. Ball et al. (2019) and Robson and Reed (1999)), there are several challenges specific to applications in estuarine regions that make this a unique category of problem. Primary amongst these is that

estuarine floods have the potential to be caused by several separate but physically connected processes, including high water levels from the ocean resulting from storm surge and/or high astronomical tide, and riverine floods due to intense 'flood-producing' rainfall in the contributing catchments (Couasnon et al., 2020; IPCC, 2012; Leonard et al., 2014; Zscheischler et al., 2018). In addition, many estuaries around the world and their contributing catchments have exhibited substantial changes in land use (e.g. urbanisation, agricultural expansion), channel modification (dredging, straightening and damming), coastal engineering works and various other modifications (Climate Change Risks to Coastal Buildings and Infrastructure, 2011; Habete and Ferreira, 2017; Hallegatte et al., 2013), with the implication that historical flood records may provide a poor guide to future hazard and risk (Milly et al., 2008; Razavi et al., 2020). Climate change adds a further layer of complexity, resulting in increasing ocean levels, changes to storm dynamics that in turn will lead to changes in both storm surges and rainfall patterns (Lowe and Gregory, 2005; Wasko and Sharma, 2015; Westra et al., 2014) as well as their dependence (Ganguli and Merz, 2019; Wahl et al., 2015; Wu and Leonard, 2019). The combination of these factors means that conventional approaches for flood risk estimation as commonly applied to inland catchments are rarely suitable for estuarine situations (Couasnon et al., 2020; Zscheischler et al., 2018).

To illustrate these challenges, consider Typhoon Rammasun, in which intense rainfall combined with storm surge produced a compound flood. As one of only two Category 5 super typhoons recorded in the South China Sea, Rammasun made landfall at its peak intensity over the island province of Hainan in China on 18[th] July 2014. It brought both heavy rainfall and strong surge with return periods of more than 100 years to the City of Haiko, the capital of Hainan province located on the estuary of Nandu River (Xu et al., 2018). Heavy rain caused widespread flooding in Haiko City and nearby urban areas. Storm surge over three meters was observed on the northern coast of the island, which prevented water from the Nandu River from draining into the sea, further exacerbating the impacts of floods in and nearby Haiko City (Wang et al., 2017). Yet flood estimation in this region proved problematic (Wang et al., 2017; Xu et al., 2018): historical flood records are short, the region has experienced rapid and extensive urbanisation including significant hydraulic changes in Nandu River leading to non-stationarity, and climate change is already modifying key flood-generating processes such as mean sea level and heavy rainfall (IPCC, 2012). This is not an isolated example; with large human populations situated at low elevations in close proximity to where rivers meet the ocean, there are many cases where interacting processes lead to complex flood dynamics and substantial impacts (e.g. Hanson et al. (2011) and Couasnon et al. (2020)). On top of this, recent studies show that the joint probability of flood drivers in estuarine areas is affected by low-frequency climate variability, such as due to the El Niño Southern Oscillation (Wu and Leonard, 2019) and may also be experiencing long-term changes (Arns et al., 2020; Bevacqua et al., 2019), making it a more challenging task to estimate future flood risk in these areas.

A generalised schematic for how the flood-producing processes interact in an estuarine region is provided in Figure 1. Conceptually, elevated estuarine water levels are often represented as the combined effect of two separate mechanisms. The first mechanism arises from extensive rainfall occurring in the upstream catchments, leading to elevated riverine flows and high water levels in the lower catchment reaches. The magnitude, timing and duration of the ensuing flood wave driven by this mechanism depends on a combination of meteorological factors (e.g. intensity, duration and spatial extent of the 'flood-producing' rainfall event) and catchment attributes (e.g. size, topography, the wetness of the catchment prior to the 'flood-producing' rainfall event, and other factors influencing the rainfall-runoff relationship). The second mechanism arises through the combination of astronomic

tides and a set of meteorological processes (e.g. tropical or extra-tropical cyclones) that produce on-shore winds and an inverse barometric effect, which in turn leads to storm surges and strong waves. The magnitude, timing and duration of elevated oceanic water levels due to this mechanism depends on the dynamics (e.g. timing and duration) of the storm surge, its superposition on the astronomic tide (i.e. the interaction of surge and tide, with

80    the greatest effects during 'spring tides' (Cowell and Thom, 1995)), and various bathymetric effects that influence propagation of the flood wave up the estuary (Resio and Westerink, 2008; Wu et al., 2017).

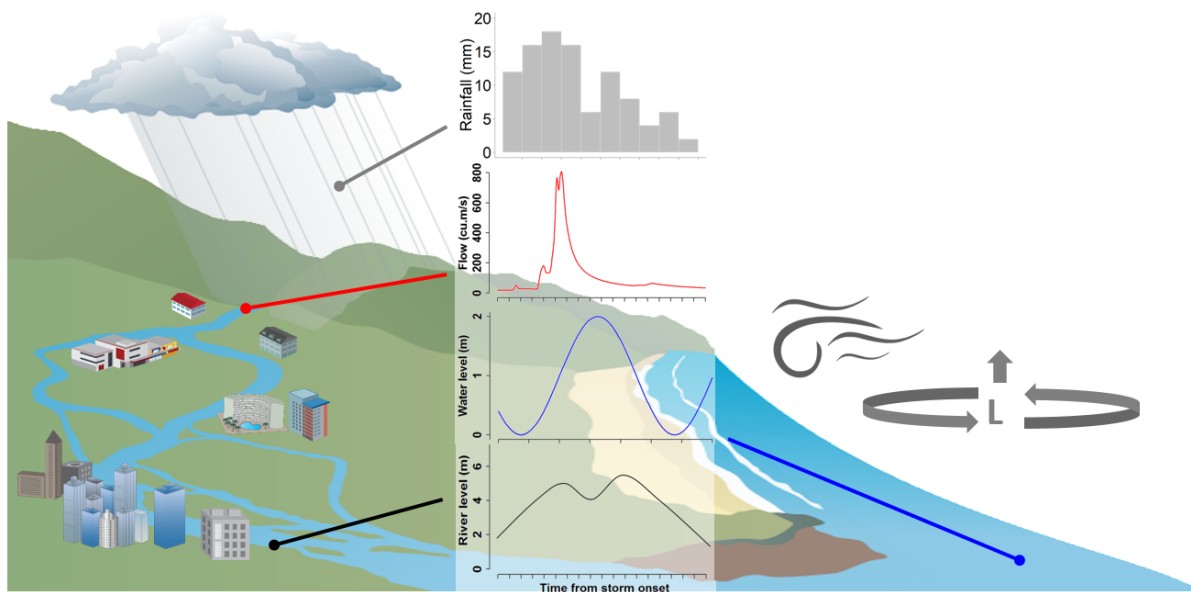

**Figure 1 Processes that commonly lead to flooding in estuarine regions with common meteorological drivers such as**
85    **wind and the inverse barometric effect. Extreme rainfall can cause significant streamflow events in upstream or local urban regions, which may combine with elevated ocean levels at the lower estuarine boundary. The specific flood magnitude depends on the timing and magnitude of constituent processes.**

Although these two physical processes are often treated separately, the flood level within an estuary is not a simple
90    addition of a fluvial hydrograph and an elevated coastal water level (Bilskie and Hagen, 2018; Ikeuchi et al., 2017; Santiago-Collazo et al., 2019). In particular, complex estuarine hydrodynamics need to be considered, and the potential for co-incident or offset timing of each component (in terms of the coincidence between the arrival of the hydrograph peak, the storm surge peak and the interaction with tidal cycles) can add considerable complexity to probability calculations. Furthermore, the meteorological drivers are sometimes (but not always) common
95    between heavy rainfall events and storm surges, such that the catchment and oceanic processes that drive estuarine floods can exhibit a non-negligible probability of occurring simultaneously (Bevacqua et al., 2017; Leonard et al., 2014; Wahl et al., 2015; Wu et al., 2018; Zheng et al., 2015a; Zscheischler et al., 2018). Methods have only started to be developed relatively recently that explicitly address this 'compounding' behaviour (Zscheischler et al., 2020).

To address this complexity and provide credible estimates of flood probability in estuarine regions, it is necessary
100    to make methodological decisions based on factors including:

- the dominant processes that have the greatest potential to produce estuarine flooding;

- the extent to which key coastal, estuarine and/or catchment properties (e.g. land use change and hydraulic structures) have changed or are anticipated to change in the future;
- the extent to which key meteorological and climatic drivers have changed or are anticipated to change in the future;
- the availability of data on either historical flooding in the estuary and/or data on the dominant flood drivers; and
- a range of other factors (e.g. availability of numerical models, methodological expectations articulated in engineering guidance documents, available budget) that ultimately will have a significant bearing on method selection.

The purpose of this paper is to provide a detailed conceptual overview of the broad approaches for estimating the probability of compound floods in estuarine regions, and review a set of specific methods available from each approach, given availability of data, calibrated models and computational power. Advantages and disadvantages of a subset of these methods are then illustrated using a real-world case study of an estuarine river system in Australia.

The rest of the paper is organised as follows. A typology of three approaches for estimating the probability of flood in estuarine regions is provided in section 2. A description of the case study area and data used in this study is provided in section 3. Details a set of specific methods selected from the three approaches and how they are applied to the case study are provided in section 4. The flood estimates produced by applying the selected methods to the case study are summarised in section 5. The discussion of main findings is included in section 6, followed by conclusions in section 7.

## 2 A Typology of Approaches for Estimating the Probability of Estuarine Floods

### 2.1 Background

A typology of different approaches for estimating estuarine flood probability is given in Figure 2. Given the requirement for probability estimation, common to all approaches is the use of a probability distribution (often, but not always, an extreme value distribution) to convert historical and/or simulated flood records or their drivers into an exceedance probability. In defining the typology, three general approaches for the probability calculation have been identified and considered here:

Approach 1: univariate flood frequency analysis applied directly to observed compound flood data;

Approach 2: univariate flood frequency analysis applied to simulated compound flood data; and

Approach 3: multivariate frequency analysis applied to key compound flood generating processes.

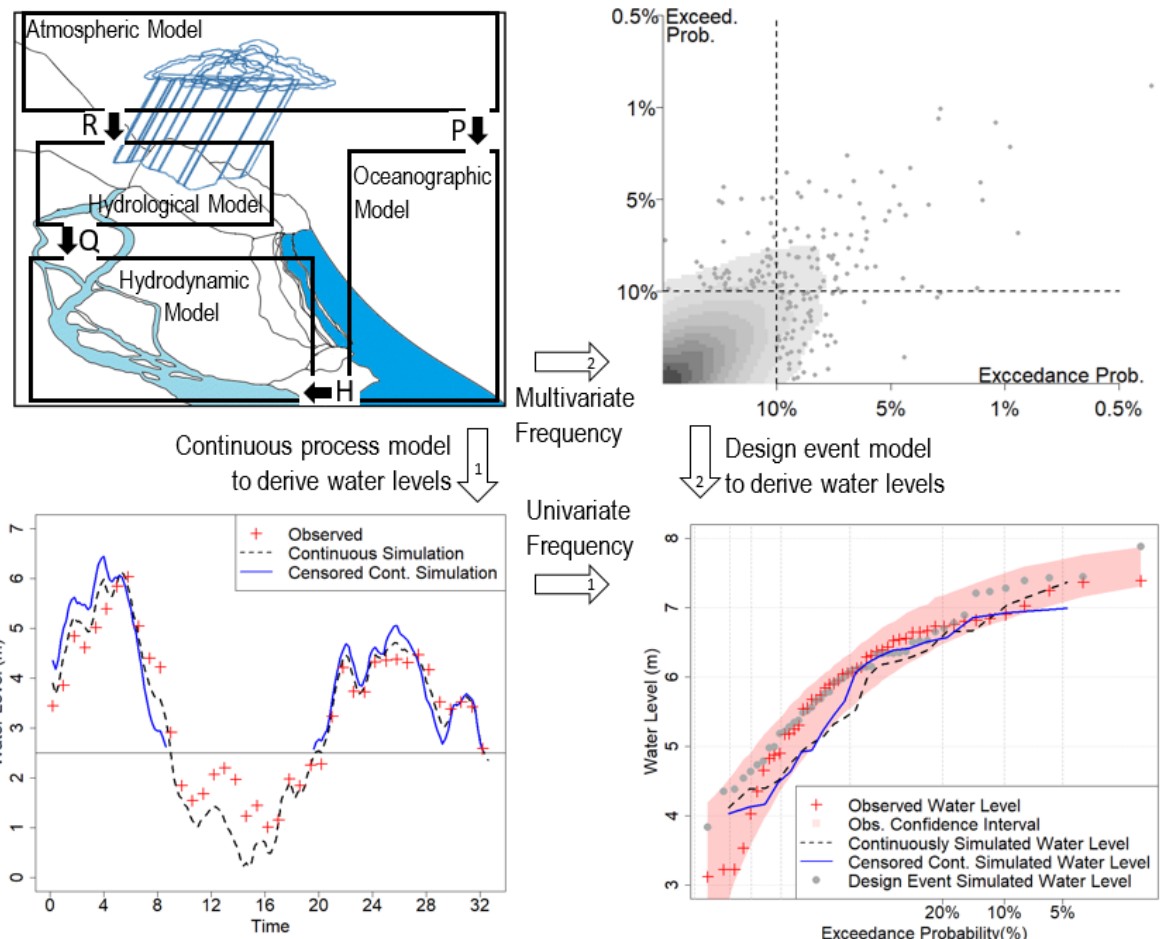

**Figure 2 Pathways for relating process modelling and statistical modelling to determine extremal water levels in estuarine river reaches, where the top left panel shows typical system boundaries for identifying relevant modelling domains (atmospheric, hydrological, oceanographic and riverine hydrodynamic) as well as key variables crossing between model domains (R – rainfall, P – pressure, W – wind, Q – streamflow, H – ocean height). Pathway 1: First transform variables to water level via continuous time-stepping process models and then apply univariate frequency analysis. Pathway 2: First abstract the system to multivariate events represented via multivariate frequency analysis, then apply design event process model to derive the compound flood water levels and their corresponding probability of exceedance.**

These approaches are defined by two key methodological decisions. The first decision is the extent to which key processes need to be explicitly resolved through numerical models, or are embedded as stationary 'boundary conditions'. In the first approach (i.e. univariate flood frequency analysis applied to observed flood data), all the physical processes that have led to the historical flood record are embedded in the observed flood data, and thus no physical modelling is required. In contrast, the remaining approaches all involve some level of numerical or statistical modelling of the key physical processes that lead to flooding, albeit with significant differences in the specific models used to implement the approaches, and the manner in which they are combined. Each of the modelling approaches therefore requires identification of a modelling domain and a set of 'boundary conditions' that delineate this domain (top left panel of Figure 2). These boundary conditions may trace back to the meteorological drivers (e.g. barometric pressure and wind data that would inform ocean models such as ROMS (Shchepetkin and McWilliams, 2005); or rainfall data that would inform hydrological models to convert rainfall

to flow), or to some intermediate variable(s) such as the historical ocean levels and/or historical fluvial flows that represent inflows to the estuary.

The second decision is the point at which a probability model is applied (i.e. directly to the variable of interest, such as flood height at a critical location, or to the drivers of flooding some distance up a modelling chain). Approaches 1 and 2 both apply a univariate probability model directly to the flood data (e.g. flood level) at the location of interest, the difference between them being whether the probability model is applied to observed historical data (Approach 1) or numerically simulated flood data (Approach 2). The univariate probability calculation is illustrated in Figure 2 by moving from the bottom left panel to the bottom right panel. Approach 2 requires the additional step of using continuous or censored continuous simulation models to move from the top left panel of Figure 2 (describing the physical processes to be simulated) to the bottom left panel (providing the continuous or censored continuous sequences of flood levels or similar flood metrics), before conducting the univariate probability calculation. In contrast, Approach 3 applies multivariate probability approaches further up the modelling chain to define multivariate 'design events' (shifting from top left to top right panel in Figure 2), which are then converted to flood levels by dynamically modelling the individual multivariate 'design events' (top right to bottom right in Figure 2).

The three primary approaches are described further in the sections below. Within each approach there is significant variety in terms of specific methods and modelling assumptions used, and a detailed review is provided for alternative implementations for each approach.

**2.2 Approach 1: Univariate flood frequency analysis applied to observed flood data**

Arguably the simplest approach is the application of a univariate probability model to observed historical flood data at the location of interest. This method is well developed (Robson and Reed, 1999) and requires sufficient historical data (to ensure sufficient accuracy in flood estimates, with a typical rule-of-thumb being the requirement of at least 30 years to estimate flood levels corresponding to probabilities up to the 1% annual exceedance probability (Ball et al., 2019)). Once this data is obtained, a univariate probability model is applied, usually to annual maxima or block maxima time series of water levels (Bezak et al., 2014; Machado et al., 2015; Wright et al., 2020). As such there is no explicit physical modelling of any constituent processes; rather, all the physical processes are considered to be embedded in the observed historical flood data.

A key assumption is that the physical 'generating processes' that gave rise to this historical record of flooding will continue into future floods (in a statistical sense), so that the probability distribution fitted to the historical data can be assumed to be stationary. Although there are many benefits to this approach—including its simplicity and transparency—there are a number of limitations:

- Historical gauges are rarely available precisely at the location(s) of interest within an estuary, with the complexity of flood wave attenuation throughout estuarine systems making it problematic to simply extrapolate information from one location to the next without consideration of the hydrodynamic processes. The lack of gauges within estuaries are likely to be at least in part due to the fact that there has historically been greater interest in measuring either the sea level or the river discharge and therefore there is less interest to place stations at the interface between the two (Bevacqua et al., 2017).

- Frequency approaches are more commonly applied to flood volume (i.e. flow) data rather than flood water level data, which can be problematic in estuarine regions where flows can be bidirectional and water levels are influenced by both upstream and downstream processes.

- Complex bathymetry and other physical features of estuarine flooding make it difficult to extrapolate the frequency curve when using observed historical records to estimate rare design events that are greater than the largest observed flood.

- Historical and/or future changes to either the estuary itself (e.g. changes to bathymetry due to dredging, coastal engineering works, natural littoral drift and fluvial sediment transport processes) and/or the upstream catchment (e.g. urbanization, agricultural expansion, reservoir construction, channel modification) can mean that historical flood record may be a poor guide to future flood probabilities.

- Historical and/or future changes to the atmospheric and oceanic drivers of flooding due to climate change, including sea level rise, storm surge and changes to rainfall patterns, can also result in the historical record being a poor guide to future flooding.

As a result of these limitations, traditional univariate flood frequency analyses applied to observed historical flood data are rarely directly appropriate for estimates of future probabilities of estuarine flooding (Yu et al., 2019), and thus one of the alternative approaches outlined below will be required for most real-world applications. Note that in situations where historical records of estuarine flooding levels are available, these data are still likely to be highly valuable to help calibrate numerical models and/or otherwise benchmark probability calculations.

## 2.3 Approach 2: Univariate flood frequency analysis applied to simulated flood data

The second approach (tracing from top left to bottom left and then to bottom right panels in Figure 2) is often referred to as 'continuous simulation', and involves simulating the dynamical flood response to continuous time series of the modelling boundary conditions using process-based models (Boughton and Droop, 2003; Sopelana et al., 2018). For example, if extended continuous historical data of catchment inflows (upper boundary condition) and ocean levels (lower boundary condition) are available, then it becomes possible to run a hydrodynamic model forced by those conditions to achieve continuous water level time series at all relevant locations within the estuary. This in turn can form the basis of a univariate flood frequency analysis applied to the simulated flood level data at the location(s) of interest. An advantage of this approach is that flood levels can be calculated at all desired locations throughout the estuary, and that changes within the estuary (e.g. changes in bathymetry, engineering works) can be explicitly captured in the model. However, the approach assumes that the physical 'generating processes' that lead to the boundary conditions are and will continue to be stationary, which is increasingly unlikely to be valid for a range of applications.

A possible solution for addressing boundary condition non-stationarity is to widen the modelling chain, thereby explicitly representing a broader range of physical processes in the model (Heavens, 2013). For example, land-use change or the construction of a reservoir in the upstream catchment can lead to significant non-stationarity in streamflow time series (the upper boundary condition in the preceding example), and this could be addressed by extending the boundary condition further up to time series of historical rainfall (Hasan et al., 2019). From there it becomes possible to explicitly model the key flow-generation processes (including the effects of land-use change and/or reservoirs) before coupling this to a hydrodynamic model of the estuary. This would enable continuous

flood height data in the estuary to be generated based on current or future catchment conditions (which would need to be parameterized into the hydrological and hydraulic models), forced in this case by historical rainfall time series. Although this approach explicitly addresses some sources of non-stationarity, evidence of climate change shifting both rainfall patterns and storm surge patterns (Lowe and Gregory, 2005; Wasko and Sharma, 2015; Westra et al., 2014) means that the assumption of stationary meteorological forcing is also increasingly questionable. Addressing this issue would lead to further widening of the boundary conditions. This is represented as ever larger boxes in the top left panel of Figure 2, defining the components of the system to be modelled and the boundary conditions to those models. Widening the modelling chain to explicitly represent an ever-increasing set of time-varying processes is certainly an attractive means to explicitly address non-stationarity of key flood generating processes. This is especially the case considering that some datasets from climate models already exist as boundary conditions for hydrodynamical modelling runs (e.g. Kanamitsu et al. (2002) and Naughton (2016)), which are helpful to assess climate change impact on compound flooding with Approach 2. However, it is important to recognise that widening the modelling chain can also lead to evermore complex models, with greater possibility of inducing biases and other forms of modelling errors into the results (Zaehle et al., 2011). This is particularly the case for climate model outputs, with the lack of hydrological validity of precipitation fields from climate models often leading to the requirement for significant bias correction or other forms of post-processing (e.g. Nahar et al. (2017)).

Furthermore, in the context of estuarine applications, the implications of anthropogenic climate change mean that it may be necessary to explicitly resolve the multivariate meteorological forcing variables that drive estuarine floods. Yet very little research has been conducted on the generation of continuous multivariate meteorological forcing variables for estuarine catchments while preserving the interactions between these variables (e.g. the joint probability of extreme rainfall and the meteorological drivers of storm surge such as pressure and wind) and eliminating their respective biases. Although approximate approaches may be available in certain instances (e.g. manually scaling the rainfall or storm surge boundary conditions), the complexity of possible future changes (e.g. heavy rainfall events being more likely to coincide with storm surge events in the future, see Seneviratne et al. (2012) and Bevacqua et al. (2019)) could render simple scaling approaches invalid. Therefore, many aspects of how to correctly apply continuous simulation approaches to estuarine floods remains an open research question.

### 2.4 Approach 3: Multivariate frequency analysis applied to key flood generating processes

The third approach involves the application of multivariate probability distributions, and is often referred to as 'event-based' because of the emphasis on deriving a series of multivariate 'design events' for further simulation through a modelling chain. These approaches are the multivariate analogy of applying IFD curves for delineating design rainfall 'events' with pre-defined probabilities, which are then converted into streamflow events that are assumed to have equivalent probability to the driving rainfall event.

These methods factorise the flood estimation problem into two separate components:

1) the estimation of a multivariate (commonly bivariate) probability distribution function based on the continuous boundary conditions; and
2) the estimation of the flood magnitude (i.e. water levels) for each combination of boundary conditions, using what is often referred to as a 'structure variable' or 'boundary function'.

A range of multivariate approaches have been applied to compound flood estimation problems, including Vine copula (Bevacqua et al., 2017), standard copulas (Muñoz et al., 2020), unit Fréchet transformations (Zheng et al., 2014), regression type models (Serafin et al., 2019) and conditional exceedance models (Jane et al., 2020). The use of copulas or equivalent formulations (e.g. unit Fréchet transformations) enables the factorisation of multivariate distributions into a set of marginal distributions and a dependence structure (i.e. a joint probability distribution). This joint probability distribution captures the defining features of the variables of interest and their interaction. For example, in Australia, a bivariate logistic extreme value distribution has been fitted to tide (observed and simulated) and rainfall data throughout the Australian coastline, and the dependence parameter of this distribution has been made available to flood practitioners across the entire coastline to describe the dependence between storm tide levels and extreme rainfall (Wu et al., 2018; Zheng et al., 2014). To capture the full joint distribution (including both marginal distributions), the dependence parameter can be coupled with publicly available IFD curves that capture the rainfall exceedance probabilities of equivalent durations, and with a frequency analysis of storm tide to reflect the lower boundary condition (Ball et al., 2019). Similar approaches exist elsewhere (e.g. Bevacqua et al. (2017), Zellou and Rahali (2019) and Moftakhari et al. (2019)), and methods are available to estimate all the key parameters of a suitable distribution when the relevant parameters are unavailable.

There are several advantages of taking an event-based approach. First, because of the emphasis on simulating a smaller number of significant 'design events', the computational loads are much lower than multi-year continuous simulations of hydrodynamic models. Second, because the drivers of estuarine flooding are factorised through the multivariate distribution, it becomes easier to incorporate the effects of future changes. This is particularly the case if one is able to assume that the dependencies between variables are either not greatly affected by climate change or that changes in dependencies produce second-order effects on flood probability compared to changes in the marginal distributions (Bevacqua et al., 2020). Under these conditions, the method can capitalise on published information on uplift factors to changes in the key marginal distributions (e.g. scaling factors for IDF curves, or for peak ocean levels), which are becoming increasingly commonly available as part of engineering flood guidance in many parts of the world (Wasko et al., in press). A further advantage is that under the assumption that the relative timing of different flood drivers is not considered (see discussion in the paragraph below), the flood surface produced using hydrodynamic models will not change under climate change; rather it is how the flood surface is converted into flood probability based on the dependence model that will change. Indeed, by separating the flood estimation problem into the two components indicated above (i.e. flood surface and associated probability), it could be possible under certain conditions to estimate the impact of future changes such as climate change on estuarine flooding without additional hydrodynamic simulations, simply by re-calculating the probabilities of the flood drivers and their dependence structure under changed future conditions.

Despite these advantages, there are several simplifications involved in this approach when converting continuous meteorological data into a set of multivariate 'design events', which could lead to significant misspecification of flood probability if not taken into account. This is illustrated through an analogy of the application of IFD curves to estimate design flood hydrographs, whereby the process of calculating IFD curves involves collapsing complex rainfall events into average rainfall intensities for different durations, resulting in the loss of the spatial and temporal dynamics of individual storm events. To convert IFDs into design floods, this additional temporal and spatial information of the rainfall event is then typically re-introduced through 'temporal patterns' and 'areal

reduction factors', respectively. Translating this analogy to multivariate design events for estuarine conditions, intensity-frequency relationships for storm tides are often derived from time series of daily maximum storm tide. During this process information on the temporal dynamics of storm surges and astronomical tides is discarded.

Although it may be possible to introduce this information on oceanographic temporal patterns through the use of 'basis functions' such as applied by Wu et al. (2017) or a similar approach by the UK Environment Agency (2019), a significant difficulty arises when trying to align the timing of the storm surge and astronomical tide events with the timing of the flood-producing rainfall in the upstream catchments (Santiago-Collazo et al., 2019). Indeed, this problem has not been resolved, with most current methods using a stochastic method to account for the temporal

shape of surge peaks (MacPherson et al., 2019) or taking a simplified approach such as assuming 'static' lower boundary conditions rather than explicitly resolving the tidal dynamics (Zheng et al., 2015a). The extent to which this simplification leads to mis-specified flood risk (and whether this misspecification leads to an under- or over-estimation of probabilities) is not known.

**3 Case Study and Data**

**3.1 Case study area and hydrodynamic model**

The case study is the Swan River system in the lower part of the Swan-Avon Basin in Western Australia, as shown in Figure 3. The total catchment area of the Swan-Avon River system is approximately 124,000 km$^2$, which makes it one of the largest river basins in Australia. The river system runs from the town of Coolgardie 500 km east of Perth to its outlet to the Indian Ocean at Fremantle. The catchment covers a large proportion of the south-western

region of Western Australia and consists of a wide range of hydrological regimes and land uses, including the relatively wet and forested areas of the Darling Scarp in the west, the Wheat belt in the middle and the semi-arid Goldfield region in the east. Due to its large size and hydrological complexity, there is currently no hydrological model available for the catchment. However, there are a few stream flow gauges near the outlet of the catchment but outside of the zone of tidal influence. These gauges include the Walyunga stream gauge and the Great Northern

Highway stream gauge and are shown in Figure 3.

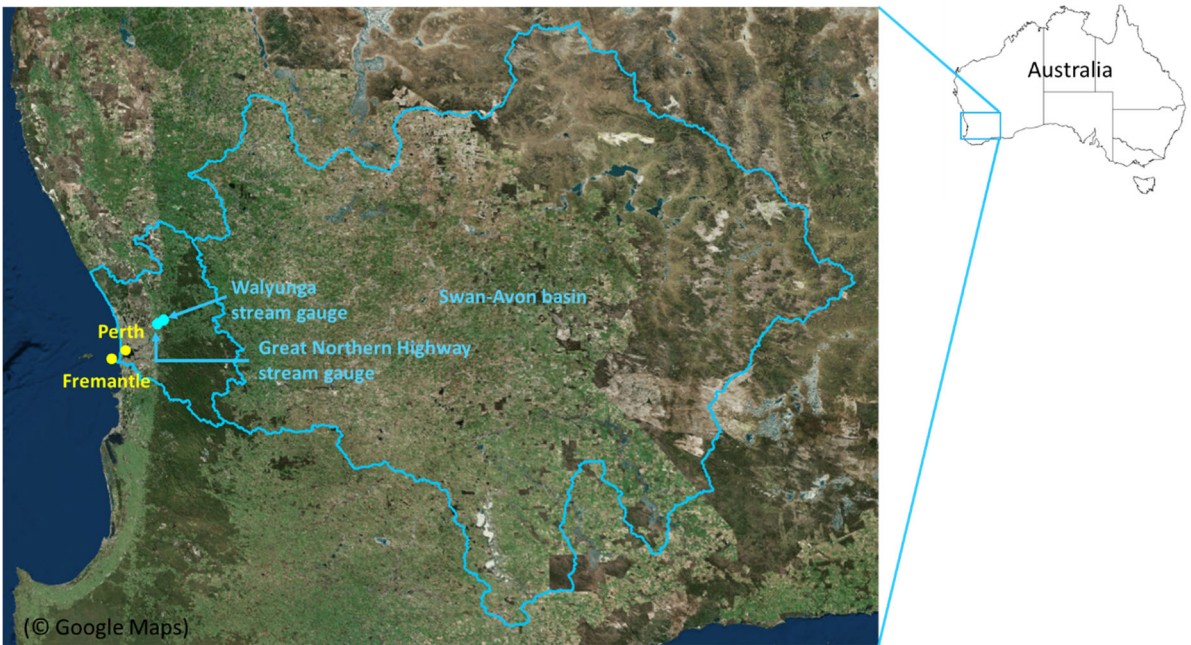

**Figure 3 Locations of Perth, Fremantle, Great Northern Highway and Walyunga stream gauges and Swan-Avon basin. The yellow dots represent the locations of major urban areas and the blue dots represent the locations of the stream gauges. (Note: This figure is created using © Google Maps.)**


The case study area is shown in Figure 4, which covers Swan River from the Great Northern Highway Bridge to its outlet at Fremantle. A two-dimensional flexible mesh hydrodynamic model is available for the study area. The model was developed using the DHI Modelling Suite MIKE21 by URS on behalf of the Department of Water and

Environmental Regulation in Western Australia to simulate water levels within the Swan and Canning Rivers' estuarine region (URS, 2013). The model domain extends from Fremantle to the Great Northern Highway Bridge 40 km north east of Perth on the Swan River, and the Pioneer Park gauge station 20 km south east of Perth on the Canning River. The main area of interest is the Swan River between Fremantle and Meadow Street Bridge, where model results are most representative of historical calibration events (URS, 2013). Therefore, 19 locations are

marked within this region and labelled from Sw1 at Fremantle to Sw19 at Meadow Street Bridge (represented by red dots in Figure 4), where flood level results are extracted from the model. The downstream boundary of the MIKE21 model is an offshore arch-shaped water level boundary located 4 km from Fremantle. The upstream boundaries are located at the Great Norther Highway Bridge on the Swan River and Pioneer Park on the Canning River. The region downstream of Sw10 is mainly storm tide dominated; the region upstream Sw16 (near the Perth

Airport) is mainly flow dominated; and the region between Sw10 and Sw16 has significant joint impact from both tail water levels at Fremantle and upstream flow, and therefore is referred to as the 'joint probability zone' or 'transition zone'.

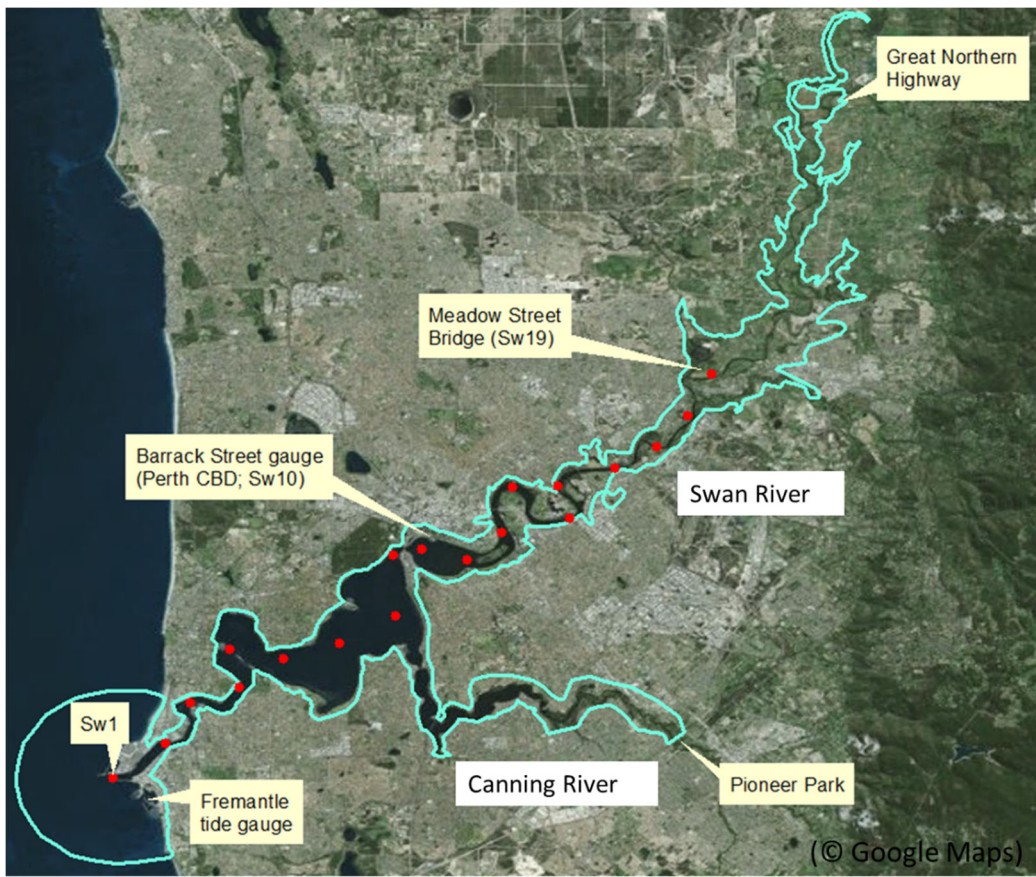

**Figure 4 Model extent and key locations for the case study system. The blue line represents hydrodynamic model extent. The red dots represent the 19 locations where flood level results are extracted, from Sw1 at Fremantle to Sw19 at Meadow Street Bridge. (Note: This figure is created using © Google Maps.)**

### 3.2 Observed data available

Water level data (i.e. not flow volume) within the estuarine regions of the Swan River is available at one gauge located at the end of Barrack Street in the City of Perth (near location Sw10 in Figure 4). The data is available from Department of Transport, Western Australia, between July 1990 and June 2015 at 15 minutes intervals with approximately 10% missing or erroneous values. This leads to about 22 years of data with no missing or erroneous values, and with water levels ranging from 0.06 m to 1.92 m.

Sea level data at Fremantle are available at hourly intervals for 118 years between 1897 and 2015 from the Bureau of Meteorology, with about 10% missing or erroneous data. The sea level data represent the combined influence of astronomical tides, storm surge and other factors that have an impact on ocean water levels, and therefore are also referred to as storm tide. The recorded sea levels range between 0.1 m and 1.95 m.

Hourly stream flow data from both the Walyunga and the Great North Highway Bridge gauge stations are obtained from the Department of Water and Environmental Regulation, Western Australia. Data from the Great North Highway Bridge gauge are available for 14 years between 1996 and 2010, which is considered to be too short for analysis of extreme events. Consequently, stream flow data from the Walyunga gauge, available between 1970 and 2016, are used. The Walyunga gauge is about 4km upstream of the Great Northern Highway Bridge, and this

distance is considered to have minimal impact on model results considering the size of the catchment. After removing missing and erroneous data, there are in total 31 years' data available. No stream flow data are available for the Canning River. This is not considered a problem, as the inflows upstream of Canning River have little impact on water levels within the study area along the Swan River (URS, 2013). Consequently, a constant small flow of 1 m$^3$/s is used as the boundary condition at Pioneer Park (URS, 2013).

## 4 Methodology

As described in section 2, each of the general approaches to the estimation of estuarine flood probabilities can be implemented in many different ways, and one specific method is applied on the real-world case study to demonstrate the advantages and disadvantages of each approach. The details of these specific methods and how they are implemented over the case study are presented in this section.

### 4.1 Method 1: Peak-over-threshold model based flood frequency analysis applied to observed flood data

Univariate flood frequency analysis is the simplest approach for estimating flood probabilities when flood data are available, and this method has been used extensively in previous studies (Guru and Jha, 2016; Seckin et al., 2014; Xu and Huang, 2011; Zhang et al., 2017). It generally involves fitting a specified distribution (e.g. Gumbel distribution, Log-Pearson Type III distribution or generalized extreme value distribution) to flood data so that the magnitude of floods can be associated with their occurrence probability (Tao and Hamed, 2000). For this study the peak-over-threshold representation of extremes is used.

The peak-over-threshold representation for extreme value analysis is based on the Pickands–Balkema–de Haan Theorem, which leads to the generalized Pareto distribution (GPD) family (Coles, 2001). Let $\{X_1, X_2 \dots, X_n\}$ be a sequence of independent and identically-distributed random variables that follow a generalized extreme value (GEV) distribution:

$$G(x) = \exp\left\{-\left[1 + \xi\left(\frac{x-\mu}{\sigma}\right)\right]^{-1/\xi}\right\} \qquad \text{Eq. 1}$$

where, $\mu, \sigma > 0$ and $\xi$ are the location, scale and shape parameters, respectively. Then, for a high threshold $u_x$, the distribution of values $Y = (X - u_x)$ conditional on $X > u_x$ converges to the GPD:

$$G(y) = 1 - \left[1 + \frac{\xi(y)}{\tilde{\sigma}}\right]^{-1/\xi} \qquad \text{Eq. 2}$$

where $y = x - u_x$ and $\tilde{\sigma} = \sigma + \xi(u - \mu)$, with $\sigma$ and $\xi$ being the scale and shape parameters of the associated GEV. Then the maximum likelihood method can be used to fit a GPD (Coles, 2001).

One challenge associated with a GPD-based frequency analysis is the choice of the threshold value u. If the threshold value is too low, it will violate the basic asymptotic assumption of the peak-over-threshold model and lead to high bias in estimation. On the other hand, if the threshold value is too high, there will be insufficient data for fitting the distribution, which can lead to high variance. The basic principal for threshold selection is to choose as low a threshold value as possible that does not invalidate the asymptotic assumption of the model. In this study, the commonly used mean residual life (MRL) plot method (Coles, 2001) is used for threshold value selection. At

the suitable threshold value, the MRL plot should be approximately linear as a function of threshold value u (Coles, 2001).

**4.2 Method 2: Peak-over-threshold model based flood frequency analysis applied to simulated flood data**

For Approach 2, univariate flood frequency analysis is applied to flood level data simulated using a 2D hydrodynamic model. To be consistent with the method selected for Approach 1, the GPD is also used. One advantage of using the peak-over-threshold model for Approach 2 is that censoring can be used to improve the efficiency of full continuous simulation using a 2D hydrodynamic model, as only values above certain high thresholds need to be included as part of the joint probability calculation. This assumption is also based on the fact that floods are relatively rare events, and therefore data from the majority of the record will not be used to estimate the probability of floods. Therefore, it is more efficient to only simulate water levels above an appropriately high threshold value, which will reduce simulation time significantly.

Censored continuous simulation for generating compound flood levels resulting from high tail water level T and large river discharge Q is illustrated in Figure 5. By selecting all of the time periods when at least one of the boundary conditions is above the pre-determined threshold, this approach aims to simulate all water levels H above a specified high threshold value. One challenge to implementing this approach is that it is not possible to know *a priori* (i.e. without simulating the full time series of joint boundary conditions) the exact value of the boundary condition thresholds that will guarantee all water levels H above the GPD threshold are simulated. For example, extreme water levels H may also be driven by non-extreme conditions of either of the flood drivers. However, the relative rareness of the extreme conditions of each flood driver and the selection of relatively low threshold values for the boundary conditions can provide reasonable assurance that flood levels above a very high threshold value required for fitting a GDP are simulated (i.e. the 'flood periods' depicted in Figure 5 always cover the periods when flood levels H are above the suitable GPD threshold value). When implementing the censored continuous simulation method, a time buffer is also defined to separate different flood periods identified. The use of a time buffer accounts for the travelling time of water in the hydrodynamic model, and further ensures that the periods when flood level H are above the suitable GPD threshold value (e.g. generated by combination of moderate flood driver levels) will be fully simulated. The combination of the flood periods and the time buffer periods is referred to as the high water level periods, when flood level time series is fully simulated using the 2D hydrodynamic model. The time periods outside these high water level periods are referred to as the 'low water level periods' and are accounted for using a resampling approach described below.


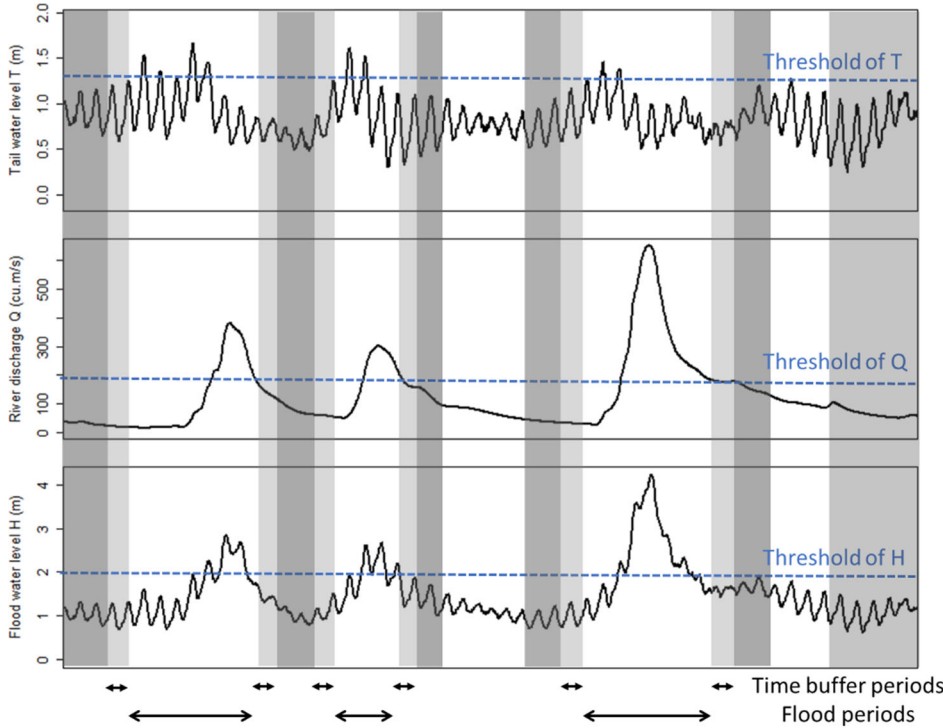

**Figure 5 Conceptual illustration of censored continuous simulation for simulating compound flood level H in estuarine regions caused by high tail water level in the ocean T and large river discharge Q. The time periods highlighted in dark grey are low water level periods; while the remaining time periods are high water level periods, which include flood periods and the time buffer.**


Since water level information below the selected threshold for fitting a GPD is censored in the frequency analysis, a resampling approach is used to fill in water level information during the low water level periods, which also addresses the challenging of not knowing *a priori* the exact value of the boundary condition thresholds. During the resampling process, a random sample of the simulation period (e.g. 1,000 hours) is selected from the original flood driver time series, subject to values of both flood drivers being below their pre-determined thresholds described above. In other words, only a fraction of the low water level periods is simulated and resampling with replacement is used to fill in flood data across the entire low water level periods. Then the corresponding flood levels are simulated using the hydrodynamic model. Thereafter, all river water level information that is not included in the high water level periods is sampled with replacement from the simulated low water level sample based on the nearest-neighbour rule applied to both the storm tide T and river flow Q values. Thus, water level information for the entire analysis period is obtained by combining the simulated water level information during the high water level periods and resampled water level information during the low water level periods.

As part of the method selected for Approach 2, the 31 years' concurrent historical sea level and river flow data are used as the basis for driving the 2D hydrodynamic model of the Swan River system. A 99th percentile threshold value is selected for both flood drivers to select flood periods for censored continuous simulation. This is equivalent to a sea water level of 1.32 m at Fremantle and a river flow of 150 m$^3$/s at the Walyunga station. A time buffer of 12 hours is selected, as the average travel time of water from the upper boundary to the lower boundary of the model is under 10 hours. In addition, a low water level period sample of 1,000 hours is randomly

selected. Thus, this process leads to a total of 29,792 hours simulation time, which is approximately 10% of the entire 31 year period under consideration. The censored simulation runs are carried out using a Windows server (with 2 × Xeon E5-2698 V3 @2.6Ghx 256 GB RAM and 2 X K80 Telsa GPU).

Once the simulated water levels are obtained, the same GPD-based frequency analysis described under Method 1 is used to estimate flood probabilities at selected locations based on these simulated water level data.

**4.3 Method 3: Event-based design variable method considering multivariate frequency analysis over key flood generating processes**

For Approach 3, the design variable method (DVM) (Zheng et al., 2015a) is selected. The DVM was initially developed as a simpler and efficient alternative to the full continuous simulation method and it includes four distinct steps: (1) event selection; (2) dependence model development; (3) flood surface simulation; and (4) final

probability estimation. The details of these four steps are described as follows.

In the first step, compound flood events caused by different flood drivers, such as storm tide and river discharge (i.e. combinations of boundary conditions with different return periods) need to be selected for simulation. Flood levels generated from these flood events will be interpolated to form flood surfaces or response surfaces with different flood magnitudes. The DVM only requires the simulation of a limited number of 'flood events' (often

on a regular grid, e.g. 10 by 10 flood events generated from combinations of flood drivers with different return levels) to produce a reasonable cover of the bivariate probability surface formed by two flood drivers (Zheng et al., 2015a; Zheng et al., 2014). In this study, both historical and synthetic flood events on an irregular grid are used to ensure flood events from drivers with significantly longer return period than the estimated flood required are included. This is recommended in order to have reasonable confidence in the estimates (Zheng et al., 2014).

In total, 28 flood events with flood drivers (i.e. storm tide and river discharge) with return periods of up to 1 in 250 years are selected based on historical record to produce a flood response surface with flood levels up to a return period of 1 in 100 years for the case study area. A summary of these flood events is provided in Table S1 in the supporting material.

In the second step, the dependence model reflecting the dependence structure between the two flood drivers and

their marginal distributions needs to be developed using either observed or simulated data (associated with component 1 of Approach 3, see section 2.4). This study follows the approach developed by Zheng et al. (2015a; 2014; 2013), where the bivariate logistic threshold excess model (Coles, 2001) is used to quantify the dependence between the two flood drivers. The model can be described using the following equation:

$$\Pr[X \leq x \cap Y \leq y] = G_{XY}(x,y) = \exp\left[-\left(\tilde{x}^{-1/\alpha} + \tilde{y}^{-1/\alpha}\right)^{\alpha}\right] \qquad \text{Eq. 3}$$

for $x > u_x$, $y > u_y$ and $0 < \alpha \leq 1$. Here, X and Y are the two stochastic variables, i.e. storm tide T and river discharge Q; x and y are realizations of X and Y; G is the bivariant distribution function of X and Y; $\tilde{x}$ and $\tilde{y}$ are the Fréchet-transformed values of x and y; $u_x$ and $u_y$ are the threshold values of x and y, above which function G is valid; and $\alpha$ is the dependence parameter, with $\alpha = 0$ representing complete dependence and $\alpha = 1$ representing complete independence. The maximum censored likelihood method can be used to estimate

parameter $\alpha$ (Tawn, 1988). For the case study, the dependence between flood drivers are estimated using observed data of storm tide and river discharge.

In the third step, the hydraulic response (i.e. simulated flood levels) of the selected flood events is simulated (associated with component 2 of Approach 3, see section 2.4). This is often done with a 2D hydrodynamic model, which can simulate the interaction between the two flood drivers. For this study, the MIKE21 model for the Swan River is used.

In the fourth and final step, the probability of different compound flood levels simulated in Step 3 can be derived based on the bivariate dependence model developed in Step 2 using the bivariate integration method introduced by Zheng (2015a). More details of this integration method can be found in Zheng et al. (2015b).

## 5 Results

The advantages and disadvantages of each approach are illustrated using the Swan River system case study. The results obtained from the specific implementation of each of the three approaches are summarised in this section.

### 5.1 Method 1

The first method based on the univariate flood frequency analysis approach is only implemented at the Barrack Street tide gauge in the City of Perth near location Sw10 in Figure 4, as this is the only location where relatively long records of observed water level data are available. The mean residual life (MRL) plot (Figure S1 in supporting material) for water levels observed at Barrack Street gauge is used for threshold selection. The mean excess stabilized around 1.37 m, which is selected to be the threshold value for fitting a GDP. The estimated return levels and their 95% confidence interval (estimated using a bootstrap method) are shown in Figure 6. The estimated flood levels range from 1.64 m for a return period of one year to 1.97 m for a return period of 200 years. The confidence intervals become increasingly wide with increasing return period, and it is important to note that return periods have been calculated based on only 22 years of historical water level data.

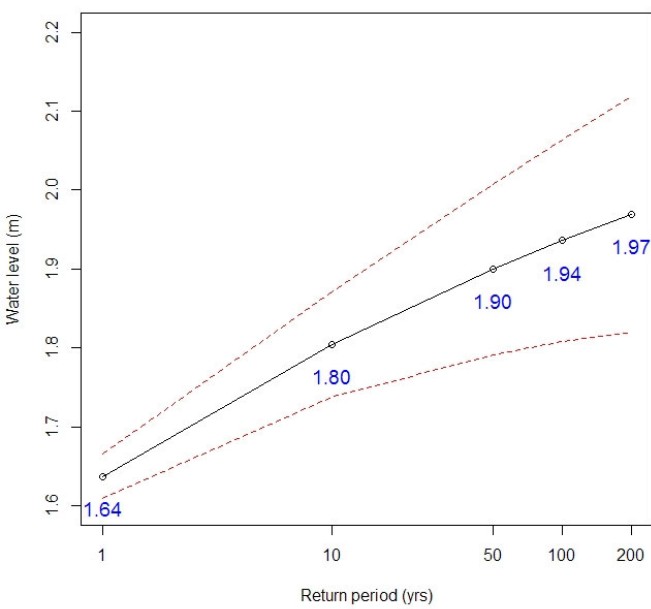

**Figure 6 Results of Method 1 applied to observed flood level data at Barrack Street gauge near location Sw10. The black line represents estimated flood levels. The red dashed lines indicate the 95% confidence interval.**


## 5.2 Method 2

For the second method adopted in this case study, hourly flood inundation data are generated using the MIKE21 model for the entire model domain for both high water level periods and the sampled low water level periods. Water level estimates from the 19 marked locations (see Figure 4) are extracted from the MIKE21 model for

analysis. Since the hourly water levels are highly correlated, the de-clustering method described in Coles (2001) is used before fitting the GPD model. In addition, the MRL plot is used to select a suitable threshold value for frequency analysis using the GPD. The MRL plots for de-clustered river level data at all 19 marked locations are provided in Figure S2 in supporting material.

In this section, results from four representative locations are selected for detailed analysis. These locations include:

location Sw1 from the tide dominated zone, locations Sw10 and Sw12 from the joint probability zone and location Sw19 from the flow dominated zone (see Figure 4). Location Sw10 is specifically selected as it is located near the Barrack Street gauge, where the only observed water level data within the river system are available (i.e. this is where the results of Method 1 and Method 2 can be directly compared). Based on the MRL plots, a threshold value of 1.3 m is selected for locations Sw1, Sw10 and Sw12; and a threshold value of 1.4 m is selected for

location Sw19.

The estimated flood levels up to a return period of 200 years and their 95% confidence intervals at these four locations are plotted in Figure 7. The results for the remaining 15 locations are provided in Figure S3 in the supporting material. The estimated return levels at Sw1, Sw10 and Sw12 are similar, with the 1 in 100 years return levels being 1.91 m, 1.89 m and 1.87 m at the three locations, respectively. The estimated 1 in 100 years flood

level at location Sw19 is much higher at 3.67 m. In addition, the 95% confidence interval for location Sw19 is much wider (higher variance) compared to the other three locations. This is mainly because location Sw19 is flow dominated and high flood levels are dominated by relatively few flood events in the historical record, leading to a more highly skewed distribution with fewer data points above the threshold for flood estimation at location Sw19 compared to the other locations.

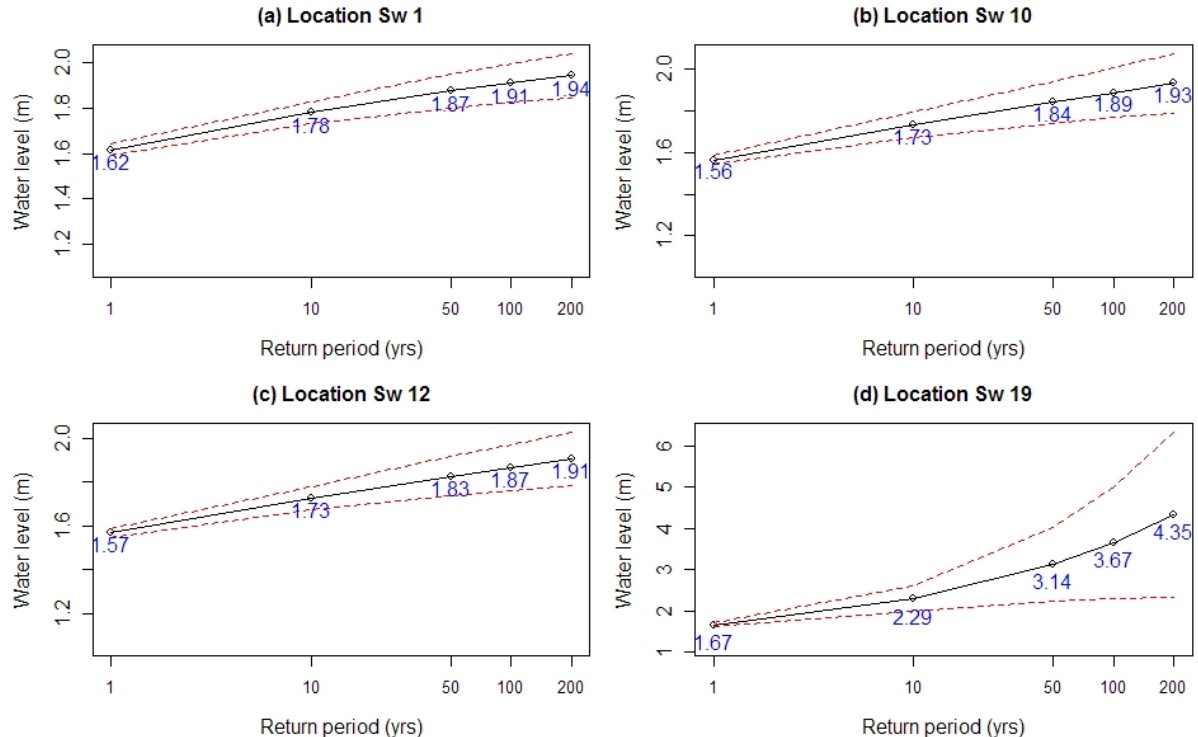


**Figure 7 Results of Method 2 applied to simulated flood level data at locations Sw1, Sw10, Sw12 and Sw19. The black lines represent estimated flood levels. The red dashed lines indicate the 95% confidence interval.**

### 5.3 Method 3

For the design variable method (DVM), the dependence between storm tide T and fluvial flood Q is first estimated using the bivariate logistic threshold excess model. The results are summarized in Figure S4 in the supporting document for a range of time lags between T and Q. The results show that the maximum dependence between storm tide T and fluvial flood Q occurs at a lag of three days with an α value of 0.88, indicating that the peak of flow often comes three days after the peak of storm tide. This lag is not surprising given that the large catchment

size generates significant lags between rainfall events (which are more likely to co-occur with the storm surge peak) and the runoff towards the catchment outlet. Therefore, an α value of 0.88 is used for flood estimation using the DVM. This is because in this method the information on the temporal dynamics of storm surges and astronomical tides is discarded and only the peaks of flood drivers and their joint dependence are considered, as discussed in section 2.4.

Flood response surfaces (i.e. flood contours) obtained for the four selected locations are presented in Figure 8. At location Sw1 where storm tide dominates the flood responses, it can be seen that as the storm tide T becomes more extreme, the flood contours become horizontal and river flow Q has little impact on flood levels. Similar phenomena can be observed for location Sw19, which is flow dominated - as river flow Q becomes more extreme (especially with a return period of 20 years or longer), flood contours become vertical and storm tide T has little

impact on resulting flood levels. In contrast, within the joint probability zone (i.e. locations Sw10 and Sw12), the flood levels are influenced by both flood drivers for the majority of the bivariate probability surface.

It can also be observed in Figure 8 that there are some variations in estimates of flood levels with very short return periods (e.g. return periods of 1 in 1 year or below), with the increase in one flood driver leading to decreased compound flood levels. Careful inspection of the results shows that this feature does not apply to any of the simulated data points, in the sense that simulation points with larger values of the boundary conditions always yield larger flood levels. Rather, the 'inflection' only occurs in a sparsely sampled region of the plot, and is thus suggestive of the limitations of using a log-linear interpolation scheme in this region. This therefore highlights the importance of carefully considering the sampling scheme as part of the analysis.


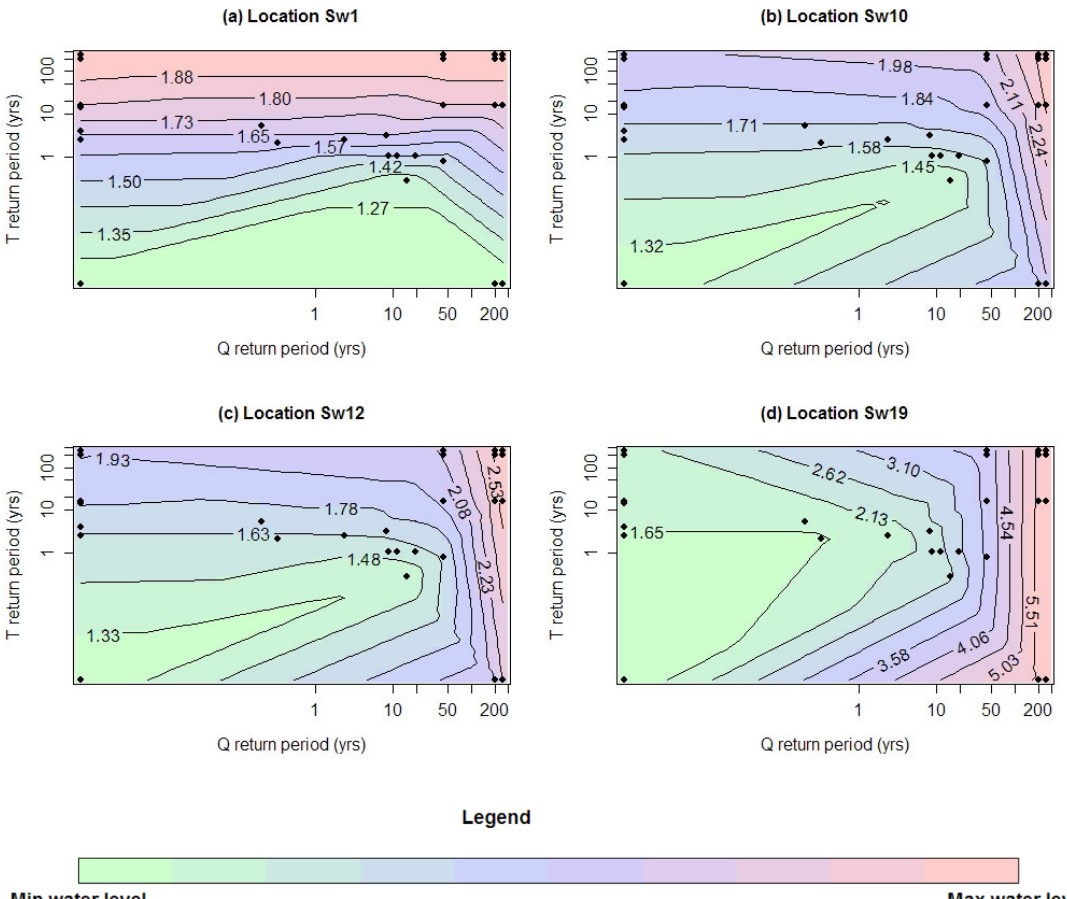

**Figure 8 Flood response surfaces (i.e. flood contours) obtained at locations Sw1, Sw10, Sw12 and Sw19. The values on the contour lines represent water levels in meters. The black dots represent the locations of the 28 flood events on the flood response surface. Note: The "inflection" in the contour lines for very short return periods is due to the use of interpolation scheme noting the sparsity of samples in these regions.**



The flood exceedance probabilities estimated using this method are plotted in Figure 9, including flood levels estimated assuming the two flood drivers are completely dependent (the red dotted lines in Figure 9), completely independent (blue dotted lines in Figure 9) and with the dependence parameter α of 0.88 (the black lines in Figure 9). As pointed out in the original study on the DVM (Zheng et al., 2015a), the maximum return period of each flood driver needs to be significantly longer than that of the response variable (i.e. flood level); therefore flood levels up to a return period of only 100 years (rather than the 200 years return period for the first two methods) are estimated here.


As shown in Figure 9, the level of dependence between the two flood drivers has little impact on the resulting flood levels at location Sw1, where water levels are dominated by storm tide. In contrast, there is a large difference in flood levels between the complete dependence and the complete independence cases in the joint probability zone (i.e. locations Sw10 and Sw12), where flood levels are determined by both tide and stream flow. Interestingly, at location Sw19 there is a large difference in flood levels resulting from the complete dependence and complete independence cases, with the largest difference of over one meter observed at a return period of 50 years. This indicates that although historically being labelled a flow-dominated zone due to high water levels being dominated by a few large riverine flood events, tidal levels also have some impact on flood levels in this area. This can also be confirmed by the results in Figure 8 that flood levels resulted from flood drivers with shorter return periods (e.g. 20 years or shorter) can be influenced by both flood drivers, although large floods at location Sw19 result predominantly from riverine flooding. These results highlight the importance of considering the dependence between all relevant flood drivers as part of the flood estimation methodology, as has been pointed out in previous studies (Moftakhari et al., 2019; Serafin et al., 2019).

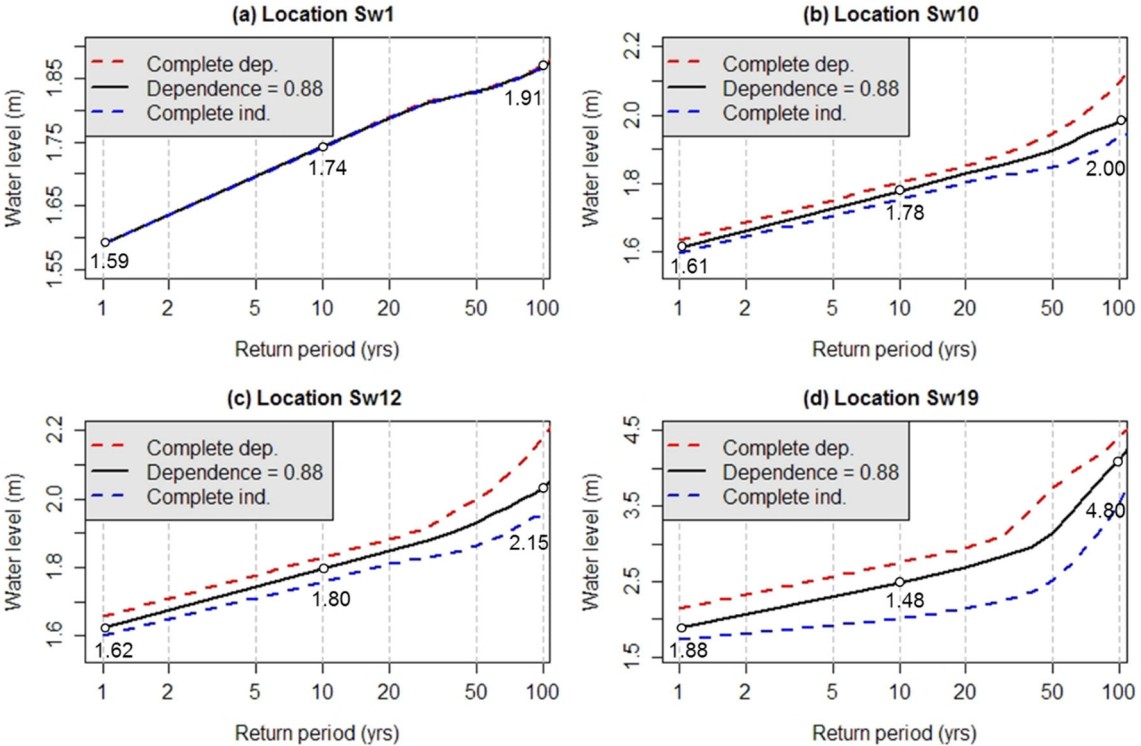

**Figure 9 Results of Method 3 applied to locations Sw1, Sw10, Sw12 and Sw19. The complete dependent and independent cases are estimated using an alpha value of 0 and 1, respectively (see section 4.3).**

## 5.4 Results comparison

A comparison between flood exceedance probabilities estimated using the three different methods is summarized in Table 1 and plotted in Figure 10. Results from Method 1 are only available at the Barrack Street gauge (near location Sw10), where observed flood data are available. Method 1 produces higher flood estimates at this location

compared to the other methods, especially for return periods of 10 years or shorter. This is very likely due to the systematic difference between the observed flood level data (with a maximum value of 1.92 m within the 22 years' data) and flood levels simulated using the MIKE21 model (with a maximum level of 1.86 m within the 31 years' analysis period) at this location. In addition, the (short) distance between the tide gauge and the modelling location Sw10 could also be a contributing factor to this difference.


**Table 1 Flood estimation results comparison**

| Loc. | Return period (yrs) | Method 1: POT[a] based FFA[b] to Observed historical data (from Approach 1) | | | Method 2: POT based FFA to simulated data (from Approach 2) | | | Method 3: DVM considering MFA to key flood drivers (from Approach 3) | | |
|---|---|---|---|---|---|---|---|---|---|---|
| | | Lower Bound (95% CI[c]) | Est. | Upper Bound (95% CI) | Lower Bound (95% CI) | Est. | Upper Bound (95% CI) | Com. Dep. | Est. | Com. Ind. |
| Sw1 | 1 | -[d] | - | - | 1.59 | 1.62 | 1.64 | 1.59 | 1.59 | 1.59 |
| | 10 | - | - | - | 1.73 | 1.78 | 1.83 | 1.74 | 1.74 | 1.74 |
| | 100 | - | - | - | 1.82 | 1.91 | 1.99 | 1.87 | 1.91 | 1.92 |
| | 200 | - | - | - | 1.85 | 1.94 | 2.04 | na[e] | na | na |
| Sw10 | 1 | 1.61 | 1.64 | 1.67 | 1.54 | 1.56 | 1.59 | 1.64 | 1.61 | 1.6 |
| | 10 | 1.74 | 1.8 | 1.87 | 1.67 | 1.73 | 1.79 | 1.8 | 1.78 | 1.75 |
| | 100 | 1.81 | 1.94 | 2.06 | 1.77 | 1.89 | 2.01 | 2.1 | 2 | 1.94 |
| | 200 | 1.82 | 1.97 | 2.12 | 1.79 | 1.93 | 2.07 | na | na | na |
| Sw12 | 1 | - | - | - | 1.55 | 1.57 | 1.59 | 1.66 | 1.62 | 1.6 |
| | 10 | - | - | - | 1.67 | 1.73 | 1.78 | 1.83 | 1.8 | 1.76 |
| | 100 | - | - | - | 1.76 | 1.87 | 1.97 | 2.18 | 2.15 | 1.98 |
| | 200 | - | - | - | 1.78 | 1.91 | 2.03 | na | na | na |
| Sw19 | 1 | - | - | - | 1.62 | 1.67 | 1.72 | 2.15 | 1.88 | 1.74 |
| | 10 | - | - | - | 1.99 | 2.29 | 2.6 | 2.75 | 2.48 | 2.01 |
| | 100 | - | - | - | 2.32 | 3.67 | 5.02 | 4.42 | 4.80 | 4.9 |
| | 200 | - | - | - | 2.35 | 4.35 | 6.35 | na | na | na |

a: POT= point-over threshold. B: FFA= flood frequency analysis c: CI = confidence interval. d: "-" indicates no data available. e: "na" indicates not applicable for extrapolation.


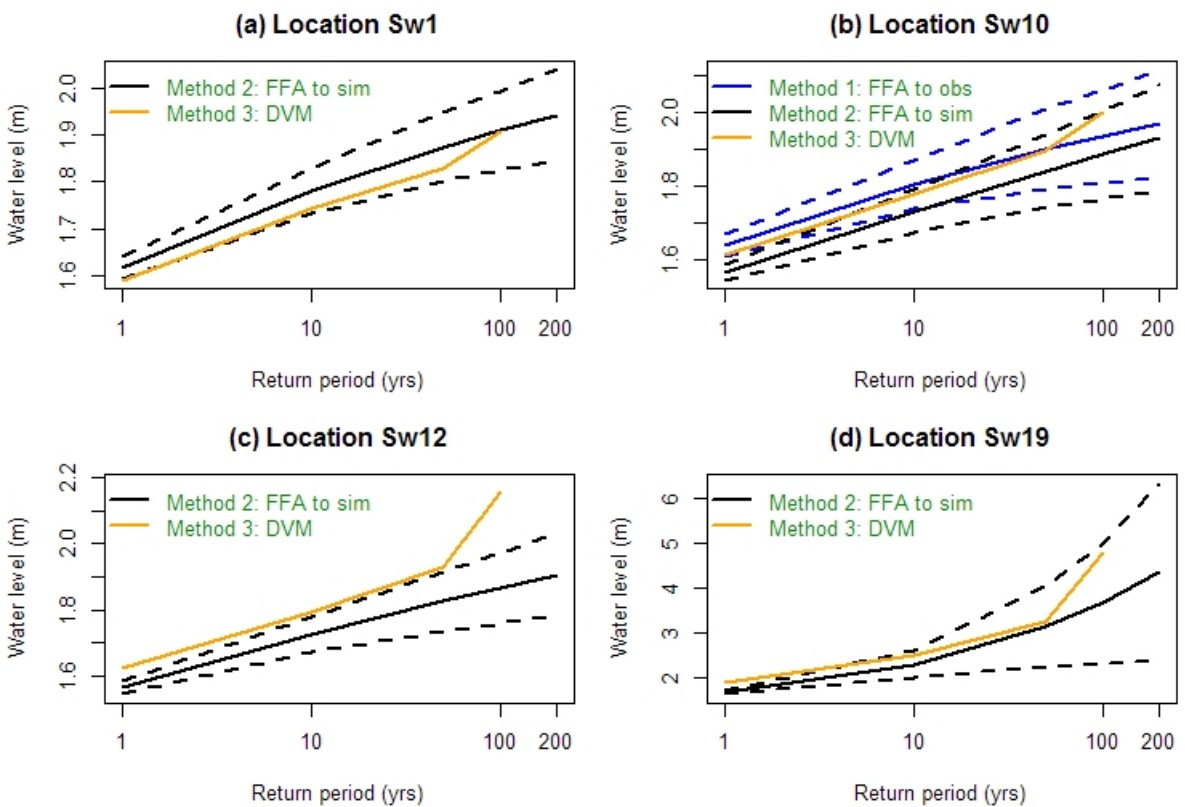

**Figure 10 Comparison between the three different methods for flood estimation. The solid lines represent estimates using each method. The dotted lines represent the 95% confidence interval where applicable.**

In regions where only one of the flood drivers dominates flood response (i.e. locations Sw1 and Sw19), Method 3 based on multivariate frequency analysis applied to flood events results in similar estimated flood levels to Method 2 based on univariate flood frequency analysis applied to simulated flood data. Estimates obtained from Method 3 are within the 95% confidence interval generated using Method 2 for most of the return periods considered. However, in the joint probability zone (e.g. locations Sw10 and Sw12) where both flood drivers have

a significant impact on resulting flood levels, the event-based Method 3 results in significantly higher flood levels for a given return period compared to Method 2. This is especially the case for location Sw12, where flood levels estimated using Method 3 are above the upper bound of the 95% confidence interval generated using Method 2 based on censored continuous simulation data. This over-estimation of flood levels for a given return period from Method 3 due to the use of a static tail water level and the associated assumption that the peaks of the two flood

drivers with always concede can potentially lead to over-conservative estimation of flood risk and costly flood prevention infrastructure.

## 6 Discussion

Each of the three approaches for flood probability estimation has their advantages and disadvantages, and these are reviewed in Table 3 and elaborated upon in the sections below.


**Table 2 Comparative summary of flood estimation approaches for estuarine floods**

| Approach | Advantages | Disadvantages |
|---|---|---|
| 1. Univariate frequency analysis applied to observed historical flood data | • Results are based directly on observed water level data (i.e. no flood modelling required).<br>• The dependence of and interactions between different flood drivers are implicitly represented within the historical water level data.<br>• Frequency analysis relies on univariate statistical theory and therefore comparatively easy to implement.<br>• Compared to multivariate methods it is easier to extrapolate to provide estimate with longer return periods. | • Long-term high-quality observed water level data is often not available.<br>• Assumes stationarity of key processes (e.g. related to hydrodynamics in the estuary or hydrology/hydraulics of the upstream catchment), which is likely to be rare in practice.<br>• Location specific, so transferability to other locations is difficult without modelling.<br>• No obvious method to incorporate the effects of climate change to estimate future flood probabilities. |
| 2. Univariate frequency analysis applied to simulated flood data | • Can be applied to entire estuarine regions.<br>• Dependence between flood drivers are taken into account implicitly based on the boundary condition data.<br>• Dynamic interactions between (i.e. the relative timing between and shapes of) flood drivers, are taken into account implicitly.<br>• Compared to multivariate methods it is easier to extrapolate to provide estimate with longer return periods.<br>• Can easily account for a large number of flood drivers (e.g. concurrent flows) in the modelling process. | • Requires long term good quality simultaneous flood driver (i.e. boundary condition) data.<br>• Relatively computational expensive, although this can be partially addressed using censored approaches.<br>• Difficult to assess future conditions, for example due to climate change, given the need to capture marginal and joint changes of the boundary conditions. |
| 3. Multivariate frequency analysis applied to selected 'flood events' | • Can be applied to entire estuarine regions.<br>• Can be used to assess future conditions with dependence model reflecting future changes without additional hydrodynamic model runs.<br>• Computationally more efficient than Approach 2, with limited flood events to be simulated. | • Dependence model between flood drivers needs to be quantified explicitly and is location-specific.<br>• Dynamic interactions between flood drivers are ignored when using static implementations such as the DVM, leading to conservative estimation of flood risk.<br>• More difficult to extrapolate for longer return periods.<br>• Generally more difficult to account for a large number of flood drivers. |

The first approach is most straight forward to apply as it does not require any additional modelling and can take into account all flood drivers and their dependence, which are implicitly represented in the observed water level

data. It is also an established approach that has been used extensively by flood researchers and practitioners. However, Approach 1 can often involve significant extrapolation, as there are often very limited observed historical flood level data available compared to the maximum return period that needs to be estimated. In this case, 22 years observed data are used to estimate flood probability up to a return period of 200 years. This leads to large uncertainty of the estimates—although for the case study presented here, the confidence intervals are

similar to the results from Approach 2 (where 31 years of boundary condition data are used). In addition, the method is restricted to the locations where the observations are recorded. Furthermore, this approach is based on the assumption of stationarity in the estuarine characteristics and associated forcing variables, which is unlikely to be true for most locations. For example, the Swan River has experienced significant changes historically, with the majority of the low-lying areas being reclaimed land (Piesse, 2017). Moreover, the estimates obtained from

historical data cannot reflect future changes in the estuarine regions.

The second approach also uses a univariate distribution, but applied to simulated water level data in the estuary. A significant advantage of this approach is that, by applying univariate frequency analysis to simulated flood level data using a 'continuous simulation' approach, flood return levels at any location within the model domain can be estimated. This approach also enables the dependence between flood drivers to be implicitly taken into account

by using concurrent historical boundary condition data that include the relevant dependencies between flood drivers. A further advantage is that there are often more long-term flood driver data (e.g. tide data and rainfall/streamflow data) than water level data in estuarine rivers, and that elements of non-stationarity (such as change to land use, hydraulic structures, bathymetry etc) can be explicitly incorporated into the modelling framework. However, depending on the nature of the models (and particularly for high-resolution hydrodynamic

models), runtime can be a significant issue, which is only partially being addressed using censored methods such as implemented in the Swan River case study. A further challenge with this method is the inclusion of climate change. In particular, given the 'continuous simulation' nature of the method, incorporation of climate change would require estimation of continuous (usually sub-daily) boundary condition time series (e.g. rainfall and storm tide) that reflect key dependence between the boundary conditions (e.g. of rainfall and the wind/pressure data that

drive storm surge). Although, these high-resolution and temporally consistent data are at present not widely available under future climate scenarios, they can potentially be developed in the future allowing Approach 2 to be used to assess compound flood probability under future changes.

The third approach based on multivariate frequency analysis applied to key flood generating processes is an efficient alternative to the traditional full continuous simulation. By separating the dependence estimation

(including marginal distribution estimation of individual flood drivers, and a dependence structure) from the flood probability estimating process, future flood probability can be estimated by updating the dependence model between flood drivers under these conditions without the requirement of additional flood simulation runs. However, by translating continuous flood time series data into a set of 'flood events', the information on coincident timing between different flood drivers is often lost, and various simplifying assumptions often need to

be made. For example, when implementing the design variable method (DVM), the tail water level is assumed to be static (i.e. no tidal dynamics) with a value that corresponds to the specified exceedance probability. This simplifies the probability estimation process by assuming that the peak of tail water will always intercept with the peak of fluvial flood at any given location within the model domain, but it ignores the dynamic interactions of the

two flood drivers, including the possibility that the peak fluvial flood wave will not occur at precisely the same time as the peak tidal cycle. Consequently, this method will always lead to over-estimation of flood levels (Zheng et al., 2015a), as have been observed from results for the case study system. Finally, other challenges with the DVM include: 1) incorporating more than two dimensions (e.g. at confluence of two rivers within an estuary) will significantly increase the complexity of the method and therefore further simplifying assumptions may be required; and 2) the dependence between the two flood drivers is location specific and needs to be estimated using an appropriate statistical model (Zheng et al., 2015a).

**7 Conclusions**

In this study, we provide a comparative review of different approaches for probability estimation of compound floods in estuarine regions. Three commonly used approaches are considered, including two approaches based on univariate frequency analysis (one applied to observed historical flood data and the other applied to simulated flood data) and one approach based on multivariate frequency analysis applied to flood drivers of selected 'flood events'. Three specific implementation methods, one from each approach, are selected and applied to a real-world estuarine system in Australia to investigate their advantages and disadvantages in the context of estimating estuarine flood probabilities. The theoretical underpinnings of the approaches, combined with findings from the case study, enable the provision of indicative guidance for selecting a suitable method for estuarine flood probability estimation, taking into account factors such as data availability, complexity of the application/analysis process, location of interest within the estuarine region, computational demands and whether or not future conditions need to be assessed.

It should be emphasised that there is no such thing as a one-size-fits-all approach. Each approach has its own advantages and disadvantages. Flood frequency analysis using observed water level data is likely to be the simplest to apply, but will only be accurate under a range of assumptions (availability of record, stationarity of key processes, etc). If these assumptions are not valid, alternative approaches including univariate frequency analysis applied to simulated (censored) continuous flood data (Approach 2), or multivariate frequency analysis applied to the boundary conditions of simulated discrete 'flood events' (Approach 3) are required. Approach 2 based on (censored) continuous simulation can fully account for the dynamic interactions between storm tide and river flow; however, it requires long term good quality data for both processes and it is relatively computational demanding. It is also difficult to be applied to assess future conditions, as new simulation models may need to be developed and simulation runs to be repeated. Approach 3 based on simulated 'flood events' is computational efficient, as only limited 'flood events' need to be simulated. It can be applied relatively easily under future conditions, as only the dependence between the flood drivers needs to be re-calculated and no additional simulation runs are required. However the inability of Approach 3 to account for the full dynamic interactions between storm tide and river flow (e.g. timing, duration, shape and their variability) in event-based simulation and the resulting simplification by using a static storm tide value will lead to conservative estimates of flood probability.

Although this study provides a comprehensive comparative reviews of the three general approaches used for flood probability estimation through the implementation of one specific method from each approach, there are a large number of alternative implementations of each approach available. Acknowledging this, further comparison

including different specific methods is required to provide a holistic picture of methods for compound flood probability estimation in estuarine regions. In addition, some of the limitations of the methods considered (e.g. the issue related to the relative timing of flood drivers and the resulting simplification for the event-based method) requires further investigation and can potentially be improved. Finally, the development of a method that can
account for a large number of flood drivers and can be easily applied under future conditions remains a research challenge.

## Data Availability

The data and hydrological models used for this study are provided by the Bureau of Meteorology in Australia, and the Department of Transport and the Department of Water and Environmental Regulation in Western Australia,
and are restricted for research purposes only. The data may be made available upon request subject to approval from corresponding departments.

## Author Contributions

All authors collaboratively designed the experiments. WW carried out the analysis. WW wrote the initial draft of the paper. All authors contributed to the subsequent editing and revision of the paper.

## Competing Interests

The authors declare that they have no conflict of interest.

## Acknowledgements

This research is funded by Australian Research Council and Western Australian Water Corporation through Linkage Project LP150100359. We also thank the Bureau of Meteorology (http://www.bom.gov.au/), the
Department of Transport (http://www.transport.wa.gov.au/) and the Department of Water and Environmental Regulation (http://www.water.wa.gov.au/) in Western Australia for providing data and models used in this study. We acknowledge DHI for providing a free MIKE FLOOD license for this project.

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
