# Peer review of "Estimating the Probability of Compound Floods in Estuarine Regions"

_Hydrology and Earth System Sciences, 2020_

## Referee Comment (RC1) · Anonymous Referee #1 · 5 Oct 2020

**Review of the paper: "Estimating the Probability of Compound Floods in Estuarine Regions Wu" by Wu et al.**

The authors provide a comparative review of three alternative approaches for assessing the local compound flooding probability. They implement the approaches for studying compound flooding in the estuary of the Swan River in Western Australia. Such an application provides a basis for discussing the advantages and limitations of the three approaches.

Overall, I did find the study very interesting and timely. I recommend to revise the manuscript based on my comments prior to publishing the manuscript.

In general, I found the introduction pleasant to read, but I would recommend improving the presentation of the methodology, especially of Method 3. In fact, I found it particularly difficult to understand some components of Method 3.

The discussion of the advantages and limitations include a part regarding the limitations/advantages that the approaches have for assessing the climate change effect on compound flooding probability. The idea of discussing this topic is certainly interesting, but it requires some revision in my view. For example, the authors mainly refer to the possibility of including changes in the dependence between the drivers through method 3, but it is not discussed the relevance of the change in the marginal distributions, which is fundamental. In particular, I understand that the authors state that method 2 is difficult to be considered for assessing climate change as it requires time series of storm tide and precipitation for the future. This would be an issue also for method 3, despite the fact that they claim that method 3 can consider climate change effects easily. See comments related on this topic below.

**Specific comments**

L47, I would cite the paper from Wahl at the end of the sentence (alreay metnioned innt he manuscript).

L64, please, consider merging this sentence with the last sentence of the previous paragraph (on the same topic).

L76 I suggest: that produce an inverse barometric effect and on-shore winds, which in turn leads to storm surges and waves

L78 water -> oceanic water level (to make clear you are referring to the sea component only in this sentence).

116, typology

128 "and considered here" after identified

L 129-130, Consider using "compound flood" here and elsewhere when referring to the compound flooding water level, such to make clear that you are referring to the resulting water level from the two drivers. For example, in the caption of Fig 2. I can certainly say that this would have made my reading easier.

L 146 "numerical" is fine here? In method 2 you may not need a numeric model (i.e., hydrodynamical model) rather use an e.g., statistical model (personally, I do not see that as numerical). I see that in your case you use numerical modelling, but this part of the manuscript appear of a more general nature.

L 166 Similarly to the above, doesn't dynamically refer to something that is not statistical? Anyway, I would modify, to make clear that such modelling can also be purely statistical.

L 159 Consider adding Approach 1,2,3, also earlier on, to give a better orientation to the reader.

L 174, do you mean 30 years of data to estimate the 1-in-100 years return level? Anyway, you may want to qualify "estimate", anything can be estimated, but would that estimate be too uncertain or not?

L 185, During a discussion among colleagues, it was hypothesised that this may be related to the fact that often there is interest in measuring either the sea level or the river discharge and therefore no stations are collocated at the interface between the two. What do you think about this? Discuss it if you think that this is relevant. I guess that this appears also discussed/hypothesised in Paprotny et al. ("Compoun flood potential in Europe").

L 188, plaase, make it clear that you are referring to the need of transforming flow into the water level

L 205, Do you have a reference? Not sure if this was given earlier.

L 227, "Although…". Consider moving this to the beginning of the next paragraph

In general, regarding sections 2.2-2.4, 230-241, I believe that the reader would benefit from finding dome additional references to works where similar approaches have been used. In my view, this can help, especially in a work like that aims at reviewing available methods.

L230-241, I find this part a bit too strong in the statements. Hydrodynamical modelling works based on oceanic and streamflow input that is available from climate models. I see that there are uncertainties and that storm tide and river flow need to be obtained based on computationally expensive modelling. But some data are available out there that can be helpful to assess the climate change impact on compound flooding even with Approach 2. I would suggest discussing this topic more.

L252, Isn't it what you produce the value of the structural variable rather than the variable itself? You use a "function" to convert a given bivariate event into a water level.

L 262, "condition." Please, provide a reference, where this method is described

L 266, how to select the design events? Multiple pairs in the bivariate space can have the same probability to occur, i.e. return period. Therefore the selection is not as easy as in the univariate case. This is discussed in the paper of Moftakhari et al. (2019). A brief discussion (2/3 sentences) on these issues is welcome.

L 268, please, clarify this sentence.

L 298 "Due to...complexity", Or is it that none has rally tried to develop one?

L368, Authors tend to oppose GPD and GEV as alternative approaches. Do you expect any differences in terms of uncertainties? Also, you use the GPD to estimate return periods/level. Shouldn't you also provide an equation for that?

L 390, Please, refer explicitly to the fact that extreme H may also be driven by non-extreme conditions of either of the drivers, therefore this should be taken into account when defining the threshold for Q and T.

L 390, It is not clear to me why you need to account for the low water level periods through the resampling approach, given that you will fit the GPD only to the extremes. I understand that is necessary to be aware of the time in between the peaks to estimate the return periods, but why simulating it?

L412, Are you simulating also a fraction only of the low water level and then using such a short simulation to fill a longer part of the time series? Please, explain better.

L 426, This is shorter than then 31years, which correspond to about 271560hours. I would highlight this explicitly as it is relevant as you implicitly suggest.

L 436, "conditions", refer to variables to guide the reader (storm tide and river discharge)

L437, Introduce the "grid" or make it clear what the grid is here in this context.

L 442, "250 years", what return period? The univariate of the individual drivers? This is unclear. Also, you may need to clarify what type of data are you using for the boundary conditions. You seem to have 22 years of data of storm tide only, how did you estimate the 250 year return period without massive uncertainties?

L 445, The dependence between the drivers within the 28 events? Please, clarify.

L458, I see that you use MIKE21 for Methods 1 and MIKE FLOOD for method 3. Can this be responsible for the differences in the results based on the two methods? Please, discuss.

L 459, step 1 was not introduced formally.

L460, this small paragraph is not clear to me. Please, explain better for people who are not familiar with the method.

472, In the methods, please mention how you retrieved the uncertainties in the estimate (based on the uncertainties in the fitted parameters).

488, I suggest to highlight that this location was used in Method 1 (so to allow for a comparison).

L 498, Aren't you also for Method 2 using the MRL plot applied to the H water level? Please specify if not done already and include a discussion on this within this sentence (one may expect that MRL to define a thresholds such result in similar uncertainties at all locations.

L 509, Why do you use the values maximising the dependence? Understanding when the dependence is maximised provides interesting information on the physical system, however, the dependence values that are relevant from a point of view of the impact is that between the variables at the same time. In fact, the storm tide and the river flow interact at the same time in the real world.

Fig 8, The 2D simulations receive as input time series of T and Q, therefore a question arises: which is the value of the time series that you consider as that to be reported on the x and y axes?

The plots, e.g., panel c, suggests that for a given 10year return level of Q, when T becomes larger (from 0 to 1-year return period), H decreases. This is physically inconsistent. Such inconsistent behaviours seem to occur in the range of T AND Q below 1-year return levels. Do you have an explanation for that? If the explanation is convincing, one would then consider not showing values in this bivariate range (up to 1-year return level for both variables).

L 524, How do you estimate the case of complete dependence/independent variables? I understand that you get the water level based on 2d simulation with input T and Q observed time series. If they were not time series, I could see the concept of independence, but in this context, I find it unclear. This comment is related to that on the general explanation of method 3.

L 550 I would reverse the sentence, highlighting the result based on method 2 and 3

compared to 1, the observation-based method. Hence, "2 and 3 lead to lower estimates than 1…"

L 565 can the comparison be affected also by the difference model type used in method 2 and 3?

Table 2,
Disadvantages for method 2, "Difficult to asses future conditions…": Why isn't this the case also for case 3?
The advantage for method 3 about future conditions: This does not seem to me as simple as stated. Please discuss. By the way, also changes in the marginals should be included, which appears to be the most relevant for future changes and at least the change for which we have the highest confidence (the confidence on the changes in the dependence is small). I would suggest dicussing this taking into account the following papers (at least):

– About changes in the dependence: Wahl et al. (Nature Climate Change)highlights a change in the dependence in the past, Bevacque et al (https://eartharxiv.org/repository/view/293/) highlights tha changes in the dependence are uncertain, and Ganguli and Merz (https://agupubs.onlinelibrary.wiley.com/doi/full/10.1029/2019GL084220) also discuss changes in the dependence for the past.

– Moftakhari et al.,(PNAS) and Bevacque et al (above) about projected changes in the marginals (i.e. Storm surge, precipitation, and sea level rise).

L 590 "stationarity", add "in the estuarine characteristics". You are referring not to the meteorological conditions here, so make it clear, please.

L598 Could you clarify/discuss why should accounting for the dependence explicitly be an advantage (compared to method 1)? Thanks.

602, Personally, I would add something along this line. "incorporated in the modelling framework", add: "through considering the most recent bathymetry characteristic of the estuary when interested in the present-day estimate of the flooding probability".

L 609, There is high-resolution data of sea level (storm surge/waves) and precipitation available, though I understand that especially for sea level, these are rare and in general can be uncertain. There are climate models. How can they be used to solve the issue? The fact that data is not widely available at high resolution does not mean, I think, that this is something to negatively judge this method given that I am not sure about what would a better alternative be.

612, "updating", Are you referring to update with respect to changes due to climate change?

If not, please discuss the climate change issue, as this is done in the other two cases.

If yes, please clarify. In addition, I do not understand how you would estimate the changes in the dependence. Do not we have the same issue as in method 2? Also, we have the problem that we need to estimate changes in the marginals, not only in the dependence. See comments above regarding this topic too.

649, See comment above about climate change. This needs to be discussed carefully.

L 655, "Implementation of each approach available" ->  "approaches available"

 Best regards.

---

## Referee Comment (RC2) · Anonymous Referee #2 · 14 Oct 2020

The manuscript introduces and discusses the strength and weaknesses of three approaches commonly applied to analyze compound flooding in estuaries. The approaches are demonstrated in the Swan River system in Western Australia where some of the discussed advantage and disadvantages manifest. The manuscript is informative, topical, and generally well written. In my opinion, after implementing the revisions given below (and those given by the other reviewer!!) the manuscript will warrant publication in HESS.

General comments.

-Since half of the manuscript is a review of the methods used in previous studies more references to previous applications of the discussed approaches would be desirable. Please see the specific comments for some examples of where this is the case.

[Figure]

-In the introduction, the two physical processes causing estuarine flooding are described in detail, however, a discussion regarding the possible mechanisms enhancing estuarine water levels due to the interaction of the two processes is missing.

-Section 2.4 would benefit from a similar brief discussion on the methods of selecting multivariate extremes perhaps a summary of Zheng et al. (2014). Also, the multivariate statistical methods used to estimate the probability of compound flood events e.g. regression type models (Serafin et al. 2019), standard copulas (Muñoz et al. 2020), Vine copula (Bevacqua et al. 2017) and conditional exceedance models (Jane et al. 2020) should so be discussed or at least listed. The selection of design events i.e. the issues with choosing hazard scenarios and the use of meta models to increase the efficiency of the numerical models also warrant a mention.

-The description of the method in Section 4.3 could be improved a lot. For instance, the link between the DVM grid and probability model is not clear to me.

Specific comments

Line 45: Wahl et al. (2015) analyzed the temporal variation in the dependence between precipitation and surge in the USA. Consider adding as a reference at the end of this sentence.

Line 48: This sentence is rather strong given that there are locations with gauges in the 'joint probably zone' and the results of a univariate probability analysis maybe satisfactory. Consider removing "if ever".

Line 90: Please consider referencing one of the many studies that have demonstrated this (see Santiago-Collazo et al. 2019).

Figure 2: This caption is the only place the word 'pathway' mentioned. Since pathway 1 concerns approach 2 and pathway 2 approach 3 consider changing the label numbers to 2 and 3 and mentioning in the caption that approach 1 just uses observational data.

Line 101: Does this not vary with distance along the channel? As stated later in the

manuscript: "The region downstream of Sw10 is mainly storm tide dominated; the region upstream Sw16 (near the Perth 320 Airport) is mainly flow dominated; and the region between Sw10 and Sw16 has significant joint impact from both tail water levels at Fremantle and upstream flow, and therefore is referred to as the 'joint probability zone'."

Line 206: Should add some examples here.

Line 244: I think the aim is to derive a series of multivariate 'design events' rather than 'translating the boundary conditions into a series of multivariate 'design events'.

Line 245: "These approaches are the multivariate analogy of applying IFD curves for delineating design rainfall 'events' with pre-defined probabilities, which are then converted into streamflow events of an equivalent probability." It is the streamflow event that corresponds to (or is associated with) the rainfall event with the predetermined probability not the streamflow events of an equivalent probability.

Line 249: Rephrase. I do not believe that "conversion" is the correct term here. The multivariate distribution describes the probability of the continuous boundary conditions.

Line 249: Also, sometimes called a "response function"! Not all the events will result in a flood. Would "flood magnitude" be more accurately termed "water level"?

Line 255: "The use of copulas or equivalent formulations (e.g. unit Fréchet transformations) enables the factorisation of multivariate distributions into a set of marginal distributions that capture the defining features of the variables of interest, together with a joint probability distribution that describes their interaction." 42 word sentence!! The joint distribution typically includes the marginal distribution and the dependence structure.

Line 268: Second, because the drivers of estuarine flooding are factorised through the multivariate distribution, it becomes easier to incorporate the effects of climate change

while preserving key dependencies between variables." This and the advantage discussed in the next sentence requires the assumption that the dependencies between the variables is stationary which should be stated. Also, "separating" maybe an easier term for readers to grasp than "factorizing" here and elsewhere.

Line 272: The downscaling approach in Bevacqua et al. (2017) which related the water level in a 'joint probability zone' to the meteorological forcing's as a way of accounting for climate change may be of interest.

Line 283: This sentence is very long and discusses two related but distinct issues. Please divide into two sentences.

Line 284: Consider adding MacPherson et al. (2019) here as another method of accounting for the temporal shape of surge peaks in stochastic modelling and Environment Agency (2019) for an example where a single shape is derived to represent the largest surge peaks at a site.

Line 287: I suggest adding a reference to a review of the numerical models used to study compound flooding by Santiago-Collazo et al. (2019) here.

Line 296: Typo. Missing an "of" after "range".

Lines 299: Grammar could be improved at the end of the sentence which starts on this line. Figure 3: Caption needs improving e.g. need to state what the colors of the points denote. Also, it is not clear why the Swan-Avon basin is split into two sections.

Line 309: If URS is an acronym it needs to be defined.

Line 321: Also commonly referred as the 'transition zone' which could be added here.

Line 331: Poor grammar. The term "good quality" is not defined, and it should be made clearer the numbers at the end of the sentence refer to water levels. Is the data missing randomly throughout the series or is there a pattern e.g. missing values only occur during storms? This should be explored.

Line 369: Is the Mrl plot method the approach used to find the GPD threshold in the other approaches listed?

Line 379: "One advantage of using the peak-over-threshold model for Approach 2 is that censoring can be used to improve the efficiency of full continuous simulation using a 2D hydrodynamic model, as only values above certain high thresholds are fully accounted for." I am a little confused here as the explanation in the introduction implies all of the water levels will be simulated when applying this approach. I appreciate the censoring is a good idea.

Line 387: "By selecting all of the time periods when at least one of the boundary conditions is above the pre-determined threshold, this approach aims to simulate all water levels H above a specified high threshold value." Moderately high values of both boundary conditions could produce high water levels above a specified highwater level threshold, but these will not be accounted for in the suggested approach.

Line 412: "a random sample of simulation period (e.g. 1,000 hours)" Is this a continuous 1,000 hour period?

Line 441: "In total, 28 flood events with flood drivers", why 28 events?

Line 459: "Finally, flood levels at the locations of interest (Step 3) are superimposed onto the bivariate dependence model 460 (Step 2) to estimate associated return periods." Not clear.

Line 524: How are the independence and full dependence return periods calculated? Once added consider rearraiging some text so that Figure 9 is discussed in the same paragraph in which it is introduced.

Line 531: Results reported in this paragraph are similar to those in Moftakhari et al. (2019) and Serafin et al. (2019) and probably elsewhere which could be cited here.

Line 551-554: "This is very likely due to the systematic difference between the observed flood level data (with a maximum value of 1.92 m within the 22 years' data) and

flood levels simulated using the MIKE21 model (with a maximum level of 1.86 m within the 31 years' analysis period) at this location." Interesting, is this due to a shortcoming of the MIKE21 model or the (short) distance between the two locations?

Line 572-574: "This over-estimation of flood levels for a given return period from Method 3 can potentially lead to over-conservative estimation of flood risk and costly flood prevention infrastructure." Or does using method 2 under-estimate flood levels and lead to under design?

Table 2: "Generally more difficult to account for a large number of flood drivers." Please expand on this. There are methods which allow the extension to more variables without restrictive assumptions regarding the nature of the dependence.

Table 2: "Can be used to assess future conditions with dependence structure reflecting future changes". Very general and also could be true for approach 2 if the continuous simulation was run for future projected climatic conditions?

Line 583: Perhaps "established" is more suitable than "well-developed"?

Line 607: "maintain key dependence between the boundary conditions". The dependence may change with time. I would highlight the fact that the method has the potential to account for climate change (unlike approach 1) as a benefit of the approach. The fact that the data is readily available is more of an (important) aside rather than a strength or limitation of the model.

Line 611-613: "By separating the dependence estimation from the flood probability estimating process, future flood probability can be estimated by updating the dependence structure between flood drivers under these conditions without the requirement of additional flood simulation runs." Again, under the assumption that the dependence structure remains stationary.

L616-622: Some mention of this should go at the end of the results section. References:

Environment Agency (2019) Coastal flood boundary conditions for the UK: update 2018 Technical summary report SC060064/TR6, Bristol, UK, pp. 19-113.

Jane, R., Cadavid, L., Obeysekera, J. & Wahl, T. 2020. Multivariate statistical modelling of the drivers of compound flood events in South Florida. Natural Hazards and Earth System Sciences, 20(10), 2681-2699.

MacPherson, L. R., Arns, A., Dangendorf, S., Vafeidis, A. T., & Jensen, J. (2019). A stochastic extreme sea level model for the German Baltic Sea coast. Journal of Geophysical Research: Oceans, 124, 2054– 2071. https://doi.org/10.1029/2018JC014718.

Muñoz, D. F., Moftakhari, H., & Moradkhani, H. (2020). Compound effects of flood drivers and wetland elevation correction on coastal flood hazard assessment. Water Resources Research, 56, e2020WR027544. https://doi.org/10.1029/2020WR027544.

Santiago-Collazo, F. L., Bilskie, M. V., & Hagen, S. C. (2019). A comprehensive review of compound inundation models in low-gradient coastal watersheds. Environmental Modelling & Software, 119, 166-181.

Serafin, K. A., Ruggiero, P., Parker, K., &Hill, D. F. (2019) What's streamflow got to do with it? A probabilistic simulation of the competing oceanographic and fluvial processes driving extreme along-river water levels, Nat. Hazards Earth Syst. Sci., 19, 1415–1431, https://doi.org/10.5194/nhess-19-1415-2019

---

## Author Comment (AC1) · 27 Nov 2020

**Responses to Reviewer 1 Comments**

**General comment 1**

The authors provide a comparative review of three alternative approaches for assessing the local compound flooding probability. They implement the approaches for studying compound flooding in the estuary of the Swan River in Western Australia. Such an application provides a basis for discussing the advantages and limitations of the three approaches. Overall, I did find the study very interesting and timely. I recommend revising the manuscript based on my comments prior to publishing the manuscript.

**Response:**

Thank you for your overall positive comments on our paper. Please find responses to your detailed comments below.

**General comment 2**

In general, I found the introduction pleasant to read, but I would recommend improving the presentation of the methodology, especially of Method 3. In fact, I found it particularly difficult to understand some components of Method 3.

**Response:**

This section on method 3 has been revised: a brief introduction has been added at the beginning to outline all of the steps involved; and the description of each step is also revised to include details of how each step is carried out (e.g. in relation to the case study used). Please also see detailed responses on relevant comments below.

**General comment 3**

The discussion of the advantages and limitations include a part regarding the limitations/advantages that the approaches have for assessing the climate change effect on compound flooding probability. The idea of discussing this topic is certainly interesting, but it requires some revision in my view. For example, the authors mainly refer to the possibility of including changes in the dependence between the drivers through method 3, but it is not discussed the relevance of the change in the marginal distributions, which is fundamental. In particular, I understand that the authors state that method 2 is difficult to be considered for assessing climate change as it requires time series of storm tide and precipitation for the future. This would be an issue also for method 3, despite the fact that they claim that method 3 can consider climate change effects easily. See comments related on this topic below.

**Response:**

We agree with the reviewer that changes in the marginal distributions as a result of climate change are of fundamental importance, and have ensured that this is clearly articulated in the manuscript.

However we maintain that there are several key advantages of using method 3 that make it much easier to apply for climate change applications in practice compared to method 2.

Of these, the most important is that the nature of the decomposition between the marginal and joint distribution in method 3 means that it is relatively straight forward to capitalise on existing information and agreed approaches (including those embedded in flood risk estimation manuals that are available in various countries around the world) that allow uplift of the 'marginal distributions'. For example in Australia, there is guidance on increasing Intensity-Duration-Frequency curves (which characterise the marginal distribution of rainfall) by 5% per degree (range 2%-10%) (Ball et al. 2019) as a result of climate change. Similar approaches to uplift factors exist in many other parts of the world. In relation to sea level rise and storm surge, one could also elect to increase the marginal distributions by 'factors' representing changes in those components. The capacity to align with standard flood estimation approaches (e.g. the Australian Rainfall and Runoff and the UK flood guidance) is more than a pragmatic advantage, as it also enables seamless interface between estuarine floods and upstream flood risk estimates, since both would be driven by the same changes to IDF curves (the same example could be used for transitioning between estuarine and coastal floods).

A secondary benefit is that, under certain conditions, the method capitalises on existing hydrodynamic model runs rather than requiring these to be repeated, since it would be possible to use changes in both the marginal and joint distribution to recalculate the probabilities of the simulated flood levels (in other words, the simulated water levels would stay the same, but the exceedance probabilities ascribed to those levels would change).

This is contrasted to method 2, which would require the full joint timeseries of the boundary conditions (e.g. sub-daily time series of wind and pressure needed as a boundary condition to storm surge, plus rainfall needed for inland catchment processes) and thus is a much more involved problem. Indeed, to our knowledge, we are not aware of any examples where this has been achieved in practice or published in the literature. Of course it could be argued that one can just apply similar scaling factors to historical time series; however in the case of rainfall, it's well known that the averages will change in a very different way to the extremes, so this scaling would be difficult in practice. As a result, solving this is a much more challenging problem and unlikely to be practically viable except for very research-heavy applications.

The following discussions (and additional references) have been added to improve clarity:

> "A range of multivariate approaches have been applied to compound flood estimation problems, including Vine copula (Bevacqua et al., 2017), standard copulas (Muñoz et al., 2020), unit Fréchet transformations (Zheng et al., 2014), regression type models (Serafin et al., 2019) and conditional exceedance models (Jane et al., 2020). The use of copulas or equivalent formulations (e.g. unit Fréchet transformations) enables the factorisation of multivariate distributions into a set of marginal distributions and a dependence structure (i.e. a joint probability distribution). This joint probability distribution captures the defining features of the variables of interest and their interaction. "

> "…,it becomes easier to incorporate the effects of future changes. This is particularly the case if one is able to assume that the dependencies between variables are either not greatly affected by climate change or that changes in dependencies produce second-order effects on flood probability compared to changes in the marginal distributions. Under these conditions, the method can capitalise on published information on uplift factors to changes

in the key marginal distributions (e.g. scaling factors for IDF curves, or for peak ocean levels), which are becoming increasingly commonly available as part of engineering flood guidance in many parts of the world (Wasko et al, in press). A further advantage is that under the assumption that the relative timing of different flood drivers is not considered (see discussion in the paragraph below), the flood surface produced using hydrodynamic models will not change under climate change; rather it is how the flood surface is converted into flood probability based on the dependence model that will change. Indeed, by separating the flood estimation problem into the two components indicated above, it could be possible under certain conditions to estimate the impact of future changes such as climate change on estuarine flooding without additional hydrodynamic simulations, simply by re-calculating the probabilities of the flood drivers and their dependence structure under changed future conditions."

In addition, the following changes have been made to section 2.4 to improve clarity:

- The 'two steps' in this section has been change to 'two components' to differentiate form the steps required when implementing each method.
- Component 2 is revised to "the estimation of the flood magnitude (i.e. water levels) for each combination of boundary conditions, using what is often referred to as a 'structure variable' or 'boundary function'."

**Specific comment 1**

L47, I would cite the paper from Wahl at the end of the sentence (already mentioned in the manuscript).

**Response:**
The reference has been added.

**Specific comment 2**

L64, please, consider merging this sentence with the last sentence of the previous paragraph (on the same topic).

**Response:**
Thank you for this suggestion. The first sentence is on the joint impact of different flood drivers. The second one is on the impact of future climate conditions on the joint impact of different flood drivers. Therefore, we felt it best that these be kept as distinct ideas.

**Specific comment 3**

L76 I suggest that produce an inverse barometric effect and on-shore winds, which in turn leads to storm surges and waves

**Response:**
Thank you for the suggestion. It is changed to "that produce on-shore winds and an inverse barometric effect, which in turn leads to storm surges and waves".

**Specific comment 4**
L78 water -> oceanic water level (to make clear you are referring to the sea component only in this sentence).

**Response:**
It has been revised as suggested.

**Specific comment 5**
L116, typology

**Response:**
Thank you for the observation. It has been corrected.

**Specific comment 6**
L128 "and considered here" after identified

**Response:**
Thank you for the suggestion. It has been added.

**Specific comment 7**
L 129-130, Consider using "compound flood" here and elsewhere when referring to the

compound flooding water level, such to make clear that you are referring to the resulting

water level from the two drivers. For example, in the caption of Fig 2. I can certainly say that

this would have made my reading easier.

**Response:**
Added, as suggested.

**Specific comment 8**

L 146 "numerical" is fine here? In method 2 you may not need a numeric model (i.e.,

hydrodynamical model) rather use an e.g., statistical model (personally, I do not see that as

numerical). I see that in your case you use numerical modelling, but this part of the

manuscript appear of a more general nature.

**Response:**

The wording has been changed to "numerical or statistical modelling".

**Specific comment 9**

L 166 Similarly to the above, doesn't dynamically refer to something that is not statistical?

Anyway, I would modify, to make clear that such modelling can also be purely statistical.

**Response:**

"Numerical modelling" is not referred here, so no changes are made.

**Specific comment 10**

L 159 Consider adding Approach 1,2,3, also earlier on, to give a better orientation to the reader.

**Response:**

Added at the beginning of section 2.1.

**Specific comment 11**

L 174, do you mean 30 years of data to estimate the 1-in-100 years return level? Anyway, you may want to qualify "estimate", anything can be estimated, but would that estimate be too uncertain or not?

**Response:**

Yes this is correct. The sentence has been revised to "to ensure sufficient accuracy in flood estimates, with a typical rule-of-thumb being the requirement of at least 30 years to estimate flood levels corresponding to probabilities up to the 1% annual exceedance probability (Ball et al., 2019)".

In addition, a statement on the uncertainty of the results obtained using this method is added in the results section:

"The confidence intervals become increasingly wide with increasing return period, and it is important to note that return periods have been calculated based on only 22 years of historical water level data."

**Specific comment 12**

L 185, During a discussion among colleagues, it was hypothesised that this may be related to the fact that often there is interest in measuring either the sea level or the river discharge and therefore no stations are collocated at the interface between the two. What do you think about this? Discuss it if you think that this is relevant. I guess that this appears also discussed/hypothesised in Paprotny et al. ("Compound flood potential in Europe").

**Response:**

Thank you for this suggestion, the following comment has been added.

"The lack of gauges within estuaries are likely to be at least in part due to the fact that there has historically been greater interest in measuring either the sea level or the river discharge and therefore there is less interest to place stations at the interface between the two (Paprotny et al., 2018)."

**Specific comment 13**

L 188, please, make it clear that you are referring to the need of transforming flow into the water level

**Response:**

The sentence has been revised to

"…, which can be problematic in estuarine regions where flows can be bidirectional and water levels are influenced by both upstream and downstream processes."

**Specific comment 14**

L 205, Do you have a reference? Not sure if this was given earlier.

**Response:**

The following references have been added:

Boughton, W. and Droop, O.: Continuous simulation for design flood estimation—a review, Environmental Modelling & Software, 18, 309-318, 2003.

Sopelana, J., Cea, L., and Ruano, S.: A continuous simulation approach for the estimation of extreme flood inundation in coastal river reaches affected by meso- and macrotides, Natural Hazards, 93, 1337-1358, 2018.

**Specific comment 15**

L 227, "Although…". Consider moving this to the beginning of the next paragraph. In general, regarding sections 2.2-2.4, 230-241, I believe that the reader would benefit from finding dome additional references to works where similar approaches have been used. In my view, this can help, especially in a work like that aims at reviewing available methods.

**Response:**

Thank you for this suggestion. The paragraph has been revised as suggested. The following references have been added.

Hasan, H. H., Mohd Razali, S. F., Ahmad Zaki, A. Z., and Mohamad Hamzah, F.: Integrated Hydrological-Hydraulic Model for Flood Simulation in Tropical Urban Catchment, Sustainability, 11, 2019.

Heavens, N. G., Ward, D. S. & Natalie, M. M. 4: Studying and Projecting Climate Change with Earth System Models., Nature Education Knowledge, 4, 4, 2013, 4(5):

Zaehle, S., Prentice, C., and Cornell, S.: The evaluation of Earth System Models: discussion summary, Procedia Environmental Sciences, 6, 216-221, 2011.

**Specific comment 16**

L230-241, I find this part a bit too strong in the statements. Hydrodynamical modelling works based on oceanic and streamflow input that is available from climate models. I see that there are uncertainties and that storm tide and river flow need to be obtained based on computationally expensive modelling. But some data are available out there that can be helpful to assess the climate change impact on compound flooding even with Approach 2. I would suggest discussing this topic more.

**Response:**

I agree that this is a topic that different people may feel differently and worth discussion. The authors believe that by extending modelling boundary the model will generally become more complex and additional errors will be introduced. It is important to recognise this challenge when considering extending model boundaries. However, it is also important to recognise that some datasets already exist as boundary conditions, which are helpful to assess the climate change impact on compound flooding even with Approach 2. This section has been revised in the manuscript:

"Widening the modelling chain to explicitly represent an ever-increasing set of time-varying processes is certainly an attractive means to explicitly address non-stationarity of key flood generating processes. This is especially the case considering that some datasets from climate models already exist as boundary conditions for hydrodynamical modelling runs (e.g. Kanamitsu et al. (2002) and Naughton (2016)), which are helpful to assess climate change impact on compound flooding with Approach 2. However, it is important to recognise that widening the modelling chain can also lead to evermore complex models, with greater possibility of inducing biases and other forms of modelling errors into the results (Zaehle et al., 2011)."

**Specific comment 17**

L252, Isn't it what you produce the value of the structural variable rather than the variable itself? You use a "function" to convert a given bivariate event into a water level.

**Response:**

It is correct that the second component of this method is to produce the water levels. This statement is revised to:

"2) the estimation of the flood magnitude (i.e. water levels) for each combination of boundary conditions, using what is often referred to as a 'structure variable' or 'boundary function'."

**Specific comment 18**

L 262, "condition." Please, provide a reference, where this method is described

**Response:**

The following reference is added.

Ball, J., Babister, M., Nathan, R., Weeks, W., Weinmann, E., Retallick, M., and Testoni, I. (Eds.): Australian Rainfall and Runoff: A Guide to Flood Estimation, Commonwealth of Australia, 2019.

**Specific comment 19**

L 266, how to select the design events? Multiple pairs in the bivariate space can have the same probability to occur, i.e. return period. Therefore the selection is not as easy as in the univariate case. This is discussed in the paper of Moftakhari et al. (2019). A brief discussion (2/3 sentences) on these issues is welcome.

**Response:**

It is true that multiple pairs of drivers in the bivariate space can have the same probability to occur. However, this is expected and not considered an issue. In fact the method is design to deal with this, as on the flood surface (e.g. Figure 8), the estimated flood contour highlights how the same flood level can occur for different combinations of both flood drivers. The key of flood event selection is to have flood drivers with a return period much longer than that of estimated flood levels, as pointed out in section 5.3.

The discussion on the selection of flood events is now included in the second paragraph of revised section 4.3.

"In the first step, compound flood events caused by different flood drivers, such as storm tide and river discharge (i.e. combinations of boundary conditions with different return periods) need to be selected for simulation. Flood levels generated from these flood events will be interpolated to form flood surfaces or response surfaces with different flood magnitudes. The DVM only requires the simulation of a limited number of 'flood events' (often on a regular grid, e.g. 10 by 10 flood events generated from combinations of flood drivers with different return levels) to produce a reasonable cover of the bivariate probability surface formed by two flood drivers (Zheng et al., 2015a; Zheng et

al., 2014). In this study, both historical and synthetic flood events on an irregular grid are used to ensure flood events from drivers with significantly longer return period than the estimated flood required are included. This is recommended in order to have reasonable confidence in the estimates (Zheng et al., 2014)."

**Specific comment 20**

L 268, please, clarify this sentence.

**Response:**

The following explanation has been added:

"This is particularly the case if one is able to assume that the dependencies between variables are either not greatly affected by climate change or that changes in dependencies produce second-order effects on flood probability compared to changes in the marginal distributions. Under these conditions, the method can capitalise on published information on uplift factors to changes in the key marginal distributions (e.g. scaling factors for IDF curves, or for peak ocean levels), which are becoming increasingly commonly available as part of engineering flood guidance in many parts of the world (Wasko et al, in press). A further advantage is that under the assumption that the relative timing of different flood drivers is not considered (see discussion in the paragraph below), the flood surface produced using hydrodynamic models will not change under climate change; rather it is how the flood surface is converted into flood probability based on the dependence model that will change."

**Specific comment 21**

L 298 "Due to...complexity", Or is it that none has rally tried to develop one?

**Response:**

This is correct. There is currently no hydrological model exist for the entire catchment, mainly due to the size and complexity of the catchment.

**Specific comment 22**

L368, Authors tend to oppose GPD and GEV as alternative approaches. Do you expect any differences in terms of uncertainties? Also, you use the GPD to estimate return periods/level. Shouldn't you also provide an equation for that?

**Response:**

The difference in the estimation outcomes from GPD vs GEV is out of the scope of this paper. The equation for the GPD is included in section 4.1.

**Specific comment 23**

L 390, Please, refer explicitly to the fact that extreme H may also be driven by non-extreme conditions of either of the drivers, therefore this should be taken into account when defining the threshold for Q and T.

**Response:**

The following additional comment has been added in the revised manuscript.

"For example, extreme water levels H may also be driven by non-extreme conditions of either of the flood drivers."

**Specific comment 24**

L 390, It is not clear to me why you need to account for the low water level periods through the resampling approach, given that you will fit the GPD only to the extremes. I understand that is necessary to be aware of the time in between the peaks to estimate the return periods, but why simulating it?

**Response:**

One important reason that flood data during low water level periods are also 'simulated' using the resampling approach is because the actual threshold values that will be used to fit the GPD is not known a priori. The resampling approach will provide a reasonable transition of flood levels between 'flood periods' and 'low water level periods' compared to just using zero values and makes sure reasonable flood level estimates will be used for flood probability estimation.

**Specific comment 25**

L412, Are you simulating also a fraction only of the low water level and then using such a short simulation to fill a longer part of the time series? Please, explain better.

**Response:**

This is correct. The following comments is added:

"In other words, only a fraction of the low water level periods is simulated and resampling with replacement is used to fill in flood data across the entire low water level periods."

**Specific comment 26**

L 426, This is shorter than then 31years, which correspond to about 271560hours. I would highlight this explicitly as it is relevant as you implicitly suggest.

**Response:**

Thank you for this suggestion. The following comment is added:

"..., which is approximately 10% the entire 31 year period under consideration."

**Specific comment 27**

L 436, "conditions", refer to variables to guide the reader (storm tide and river discharge)

**Response:**

Thank you for the suggestion. The following wording has been added:

"i.e. flood drivers, such as storm tide and river discharge"

**Specific comment 28**

L437, Introduce the "grid" or make it clear what the grid is here in this context.

**Response:**

The following wording has been added to improve clarity:

"e.g. 10 by 10 flood events generated from combinations of flood drivers with different return levels"

**Specific comment 29**

L 442, "250 years", what return period? The univariate of the individual drivers? This is unclear. Also, you may need to clarify what type of data are you using for the boundary conditions. You seem to have 22 years of data of storm tide only, how did you estimate the 250 year return period without massive uncertainties?

**Response:**

The return period has been changed to "1 in 250 years". As indicated in the sentence, the return period refers to that of flood drivers (i.e. univariate).

As described in both section 2.4 and section 4.3, Method 3 under Approach 3 is event based and no continuous flood data are used. There are in total 28 flood events are used. These events have univariate flood drivers with return periods up to 1 in 250 years, in order to estimate compound flood levels up to 1 in 100 year return period. A summary of these flood events is provided in the supporting material. The locations of these 28 event in the flood surfaces are indicated by the black dots in Figure 28.

The 22 years of data are observed water level data at a tide gauge near location Sw10. They are used for Method 1 only. Additional comment has been added in section 5.1 on results from Method 1 to emphasise this.

"The confidence intervals become increasingly wide with increasing return period, and it is important to note that return periods have been calculated based on only 22 years of historical water level data."

**Specific comment 30**

L 445, The dependence between the drivers within the 28 events? Please, clarify.

**Response:**

The dependence here refers to the dependence between the flood drivers, which are not estimated using the 28 flood events. The dependence structure between the flood divers can be estimated using observed or simulated data. In this study, it was estimated using observed flood driver data. The following comment has been added to improve clarity.

"For the case study, the dependence between flood drivers are estimated using observed data of storm tide and river discharge."

**Specific comment 31**

L458, I see that you use MIKE21 for Methods 1 and MIKE FLOOD for method 3. Can this be responsible for the differences in the results based on the two methods? Please, discuss.

**Response:**

Sorry for the confusion. MIKE21 is one module in MIKEFLOOD. The same hydrodynamic model is used for Method 2 and Method 3. MIKEFLOOD has been changed to MIKE21 in the manuscript, except in Acknowledgements, where MIKEFLOOD license is mentioned.

**Specific comment 32**

L 459, step 1 was not introduced formally.

**Response:**

Section 4.3 on Method 3 has been revised to improve clarity. The writing in this section has been changed.

**Specific comment 33**

L460, this small paragraph is not clear to me. Please, explain better for people who are not familiar with the method.

**Response:**

This is the final step of estimating flood probability by integrating the generated flood surface in step 3 and estimated dependence model in step 2 using an integration method. The details of the integration method are out of the scope of this paper. A reference has been provided for readers who are interested in the details. In addition, this paragraph has been rewritten to improve clarity.

"In the fourth and final step, the probability of different compound flood levels simulated in Step 3 can be derived based on the bivariate dependence model developed in Step 2 using the bivariate integration method introduced by Zheng (2015a). More details of this integration method can be found in Zheng et al. (2015b)."

**Specific comment 34**

472, In the methods, please mention how you retrieved the uncertainties in the estimate (based on the uncertainties in the fitted parameters).

**Response:**

The following phrase has been added to improve clarity.

"…(estimated using a bootstrap method)"

**Specific comment 35**

488, I suggest highlighting that this location was used in Method 1 (so to allow for a comparison).

**Response:**

Thank you for the suggestion. "(i.e. this is where the results of Method 1 and Method 2 can be directly compared)" has been added.

**Specific comment 36**

L 498, Aren't you also for Method 2 using the MRL plot applied to the H water level? Please specify if not done already and include a discussion on this within this sentence (one may expect that MRL to define a thresholds such result in similar uncertainties at all locations.

**Response:**

It is correct that MRL plots are used in Method 2. This part of discussion is still under Method 2.

**Specific comment 37**

L 509, Why do you use the values maximising the dependence? Understanding when the dependence is maximised provides interesting information on the physical system, however, the dependence values that are relevant from a point of view of the impact is that between the variables at the same time. In fact, the storm tide and the river flow interact at the same time in the real world.

**Response:**

This is because in Method 3 the information on the temporal dynamics (i.e. relative timing) of storm surges and astronomical tides is discarded and only the peaks of flood drivers are considered via the use of a static tail water level, as discussed in section 2.4. This is one of the limitations of Approach 3 and thus, Method 3. The following statement has been added to improve clarity:

"This is because in this method the information on the temporal dynamics of storm surges and astronomical tides is discarded and only the only the peaks of flood drivers and their joint dependence are considered, as discussed in section 2.4."

**Specific comment 38**

Fig 8, The 2D simulations receive as input time series of T and Q, therefore a question arises: which is the value of the time series that you consider as that to be reported on the x and y axes?

The plots, e.g., panel c, suggests that for a given 10 year return level of Q, when T becomes larger (from 0 to 1-year return period), H decreases. This is physically inconsistent. Such inconsistent behaviours seem to occur in the range of T AND Q below 1-year return levels. Do you have an explanation for that? If the explanation is convincing, one would then consider not showing values in this bivariate range (up to 1-year return level for both variables).

**Response:**

This variation is potentially caused by the interpolation method used. Additional discussion has been added in the revised manuscript to explain this.

"It can also be observed in **Error! Reference source not found.** that there are some variations in estimates of flood levels with very short return periods (e.g. return periods of 1 in 1 year or below), with the increase in one flood driver leading to decreased compound flood levels. Careful inspection of the results shows that this feature does not apply to any of the simulated data points, in the sense that simulation points with larger values of the boundary conditions always yield larger flood levels. Rather, the 'inflection' only occurs in a sparsely sampled region of the plot, and is thus suggestive of the limitations of using a log-linear interpolation scheme in this region. This therefore highlights the importance of carefully considering the sampling scheme as part of the analysis."

**Specific comment 39**

L 524, How do you estimate the case of complete dependence/independent variables? I understand that you get the water level based on 2d simulation with input T and Q observed time series. If they were not time series, I could see the concept of independence, but in this context, I find it unclear. This comment is related to that on the general explanation of method 3.

**Response:**

The case of complete dependence/independence can be estimated by using different alpha values representing complete dependence (i.e. alpha = 0) and complete impendence (i.e. alpha = 1). This is discussed in section 4.3. Additional reference to section 4.3 has been added here to improve clarity.

**Specific comment 40**

L 550 I would reverse the sentence, highlighting the result based on method 2 and 3 compared to 1, the observation-based method. Hence, "2 and 3 lead to lower estimates than 1…"

**Response:**

Thank you for this suggestion. As this part focuses on results of Method 1, the way the results are discussed is not changed.

**Specific comment 41**

L 565 can the comparison be affected also by the difference model type used in method 2 and 3?

**Response:**

The same model is used for Method 2 and Method 3. See response to Specific Comment 31 above.

**Specific comment 42**

Table 2,

Disadvantages for method 2, "Difficult to assess future conditions…": Why isn't this the case also for case 3?

The advantage for method 3 about future conditions: This does not seem to me as simple as stated. Please discuss. By the way, also changes in the marginals should be included, which appears to be the most relevant for future changes and at least the change for which we have the highest confidence (the confidence on the changes in the dependence is small). I would suggest discussing this taking into account the following papers (at least):

– About changes in the dependence: Wahl et al. (Nature Climate Change)highlights a

change in the dependence in the past, Bevacque et al (https://eartharxiv.org/repository/view/293/) highlights the changes in the dependence are uncertain, and Ganguli and Merz (https://agupubs.onlinelibrary.wiley.com/doi/full/10.1029/2019GL084220) also discuss changes in the dependence for the past.

– Moftakhari et al.,(PNAS) and Bevacque et al (above) about projected changes in the marginals (i.e. Storm surge, precipitation, and sea level rise).

**Response:**

For response to the comment on advantages of Method 3, please refer to response to General Comment 3 above.

In addition, the end of discussion on Approach 2 in section 6 has been revised to:

"Although, these high-resolution and temporally consistent data are at present not widely available under future climate scenarios, they can potentially be developed in the future allowing Approach 2 to be used to assess compound flood probability under future changes."

The following paper has been added in the revised manuscript.

Ganguli, P. and Merz, B.: Trends in Compound Flooding in Northwestern Europe During 1901–2014, Geophysical Research Letters, 46, 10810-10820, 2019.

The Bevacque et al. paper (https://eartharxiv.org/repository/view/293/) is not peer reviewed yet and is therefore not included.

**Specific comment 43**

L 590 "stationarity", add "in the estuarine characteristics". You are referring not to the meteorological conditions here, so make it clear, please.

**Response:**

Thank you for this suggestion. It has been changed.

**Specific comment 44**

L598 Could you clarify/discuss why should accounting for the dependence explicitly be an advantage (compared to method 1)? Thanks.

**Response:**

By accounting for dependence implicitly, the method has one less variables to estimate, which simplifies the estimation process and reduces chance of introducing additional error/uncertainty.

**Specific comment 45**

L602, Personally, I would add something along this line. "incorporated in the modelling framework", add: "through considering the most recent bathymetry characteristic of the estuary when interested in the present-day estimate of the flooding probability".

**Response:**

There are multiple ways long-term driver data can be incorporated. The authors believe that it is better to leave it open rather than suggesting a specific way how the data are incorporated. For example, in order to address future changes, future projections of changes in the estuarine regions need to be used. Therefore, no changes have been made here.

**Specific comment 46**

L 609, There is high-resolution data of sea level (storm surge/waves) and precipitation available, though I understand that especially for sea level, these are rare and in general can be uncertain. There are climate models. How can they be used to solve the issue? The fact that data is not widely available at high resolution does not mean, I think, that this is something to negatively judge this method given that I am not sure about what would a better alternative be.

Response:

The statement referred to is "These high-resolution and temporally consistent data are at present not widely available under future climate scenarios."

This statements here simply points out the fact the high resolution data required for this method may not be readily available. There is no judging of this method involved.

**Specific comment 47**

L612, "updating", Are you referring to update with respect to changes due to climate change? If not, please discuss the climate change issue, as this is done in the other two cases. If yes, please clarify. In addition, I do not understand how you would estimate the changes in the dependence. Do not we have the same issue as in method 2? Also, we have the problem that we need to estimate changes in the marginals, not only in the dependence. See comments above regarding this topic too.

Response:

The dependence model includes the marginal distributions for individual flood drivers and the dependence structure, and can be estimated under future conditions, considering but not limited to climate change. The advantage of method 3 is that the hydrodynamic runs required to produce the flood surface will not need to be repeated. For detailed response, please refer to response to General Comment 3.

**Specific comment 48**

L649, See comment above about climate change. This needs to be discussed carefully.

Response:

See response to Specific Comment 47 and General Comment 3 above.

**Specific comment 49**

L 655, "Implementation of each approach available" -> "approaches available"

Response:

This sentence emphasis that there are multiple implementations of each approach available. Therefore, it has not been changed.

---

## Author Comment (AC2) · 27 Nov 2020

**Responses to Reviewer 2 Comments**

**General comment 1**

Since half of the manuscript is a review of the methods used in previous studies more references to previous applications of the discussed approaches would be desirable. Please see the specific comments for some examples of where this is the case.

**Response:**

Thank you very much for this suggestion. Relevant references have been added at various locations. Please see responses to specific comments below.

**General comment 2**

In the introduction, the two physical processes causing estuarine flooding are described in detail, however, a discussion regarding the possible mechanisms enhancing estuarine water levels due to the interaction of the two processes is missing.

**Response:**

The discussion of the interaction of the two process and its impact on compound flood is included in the paragraph after Figure 1.

**General comment 3**

Section 2.4 would benefit from a similar brief discussion on the methods of selecting multivariate extremes perhaps a summary of Zheng et al. (2014). Also, the multivariate statistical methods used to estimate the probability of compound flood events e.g. regression type models (Serafin et al. 2019), standard copulas (Muñoz et al. 2020), Vine copula (Bevacqua et al. 2017) and conditional exceedance models (Jane et al. 2020) should so be discussed or at least listed. The selection of design events i.e. the issues with choosing hazard scenarios and the use of meta models to increase the efficiency of the numerical models also warrant a mention.

**Response:**

The method by Zheng et al (2014) is introduced in section 4.3. A discussion including the references recommend above has been added in the revised manuscript. The following additional references have been added.

Jane, R., Cadavid, L., Obeysekera, J. &Wahl, T. 2020. Multivariate statistical modelling of the drivers of compound flood events in South Florida. Natural Hazards and Earth System Sciences, 20(10), 2681-2699.
Muñoz, D. F., Moftakhari, H., & Moradkhani, H. (2020). Compound effects of flood drivers and wetland elevation correction on coastal flood hazard assessment. Water Resources Research, 56,
Serafin, K. A., Ruggiero, P., Parker, K., &Hill, D. F. (2019) What's streamflow got to do with it? A probabilistic simulation of the competing oceanographic and fluvial processes driving extreme along-river water levels, Nat. Hazards Earth Syst. Sci., 19, 1415–1431, https://doi.org/10.5194/nhess-19-1415-2019

**General comment 4**

The description of the method in Section 4.3 could be improved a lot. For instance, the link between the DVM grid and probability model is not clear to me.

Response:

This section has been revised to improve clarity.

**Specific comment 1**

Line 45: Wahl et al. (2015) analyzed the temporal variation in the dependence between precipitation and surge in the USA. Consider adding as a reference at the end of this sentence.

Response:

Thank you for this suggestion. This reference has been added at the end of this sentence.

**Specific comment 2**

Line 48: This sentence is rather strong given that there are locations with gauges in the 'joint probably zone' and the results of a univariate probability analysis maybe satisfactory. Consider removing "if ever".

Response:

"If ever" is removed as suggested.

**Specific comment 3**

Line 90: Please consider referencing one of the many studies that have demonstrated this (see Santiago-Collazo et al. 2019).

Response:

The following references have been added.

Santiago-Collazo, F. L., Bilskie, M. V., & Hagen, S. C. (2019). A comprehensive review of compound inundation models in low-gradient coastal watersheds. Environmental Modelling & Software, 119, 166-181.
Bilskie, M. V. and Hagen, S. C.: Defining Flood Zone Transitions in Low-Gradient Coastal Regions, Geophysical Research Letters, 45, 2761-2770, 2018.

Ikeuchi, H., Hirabayashi, Y., Yamazaki, D., Muis, S., Ward, P. J., Winsemius, H. C., Verlaan, M., and Kanae, S.: Compound simulation of fluvial floods and storm surges in a global coupled river-coast flood model: Model development and its application to 2007 Cyclone Sidr in Bangladesh, Journal of Advances in Modeling Earth Systems, 9, 1847-1862, 2017.

**Specific comment 4**

Figure 2: This caption is the only place the word 'pathway' mentioned. Since pathway 1 concerns approach 2 and pathway 2 approach 3 consider changing the label numbers to 2 and 3 and mentioning in the caption that approach 1 just uses observational data.

**Response:**

The pathways are related to how the dependence is estimated, e.g. via a univariate or multivariate frequency analysis, rather than the type of data being used - observed or simulated. Pathway 1 concerns both Approach 1 and Approach 2, where observed and simulated data are used respectively. Therefore, the type of data is not mentioned in the caption of this figure.

**Specific comment 5**

Line 319: Does this not vary with distance along the channel? As stated later in the manuscript: "The region downstream of Sw10 is mainly storm tide dominated; the region upstream Sw16 (near the Perth 320 Airport) is mainly flow dominated; and the region between Sw10 and Sw16 has significant joint impact from both tail water levels at Fremantle and upstream flow, and therefore is referred to as the 'joint probability zone'."

**Response:**

This does vary with the distance of the channel. However, it is also affected by the topography of area. As shown in Figure 4, there is a very narrow section of the channel right downstream of location Sw10, which has reduced the impact of the tide in regions upstream. In addition, this classification is not absolute. As can be seen in results that even at location Sw19, there will be some impact of the tide for small flood events. Therefore, in this section it is stated that "The region downstream of Sw10 is _mainly_ storm tide dominated; the region upstream Sw16 (near the Perth Airport) is _mainly_ flow dominated, …"

**Specific comment 6**

Line 206: Should add some examples here.

**Response:**

The following references have been added:

Boughton, W. and Droop, O.: Continuous simulation for design flood estimation—a review, Environmental Modelling & Software, 18, 309-318, 2003.

Sopelana, J., Cea, L., and Ruano, S.: A continuous simulation approach for the estimation of extreme flood inundation in coastal river reaches affected by meso- and macrotides, Natural Hazards, 93, 1337-1358, 2018.

**Specific comment 7**

Line 244: I think the aim is to derive a series of multivariate 'design events' rather than 'translating the boundary conditions into a series of multivariate 'design events'.

**Response:**

This sentence refers to how the dependence is estimated. This sentence has been revised to:

"..., because of the emphasis on deriving a series of multivariate 'design events' for further simulation through a modelling chain."

**Specific comment 8**

Line 245: "These approaches are the multivariate analogy of applying IFD curves for delineating design rainfall 'events' with pre-defined probabilities, which are then converted into streamflow events of an equivalent probability." It is the streamflow event that corresponds to (or is associated with) the rainfall event with the predetermined probability not the streamflow events of an equivalent probability.

Response:

This is correct. The sentence has been revised as suggested.

**Specific comment 9**

Line 249: Rephrase. I do not believe that "conversion" is the correct term here. The multivariate distribution describes the probability of the continuous boundary conditions.

Response:

The statement has been revised to

"1) the estimation of a multivariate (commonly bivariate) probability distribution function based on the continuous boundary conditions."

**Specific comment 10**

Line 249: Also, sometimes called a "response function"! Not all the events will result in a flood. Would "flood magnitude" be more accurately termed "water level"?

Response:

"Flood magnitude" has been changed to "water level" as suggested.

**Specific comment 11**

Line 255: "The use of copulas or equivalent formulations (e.g. unit Fréchet transformations) enables the factorisation of multivariate distributions into a set of marginal distributions that capture the defining features of the variables of interest, together with a joint probability distribution that describes their interaction." 42 word sentence!! The joint distribution typically includes the marginal distribution and the dependence structure.

Response:

Thank you for this suggestion. The original sentence has been changed to:

"The use of copulas or equivalent formulations (e.g. unit Fréchet transformations) enables the factorisation of multivariate distributions into a set of marginal distributions and a dependence

structure (i.e. a joint probability distribution). This joint probability distribution captures the defining features of the variables of interest and their interaction."

**Specific comment 12**

Line 268: Second, because the drivers of estuarine flooding are factorised through the multivariate distribution, it becomes easier to incorporate the effects of climate change while preserving key dependencies between variables." This and the advantage discussed in the next sentence requires the assumption that the dependencies between the variables is stationary which should be stated. Also, "separating" maybe an easier term for readers to grasp than "factorizing" here and elsewhere.

Response:

Thank you for this suggestion. This section has been revised to improve clarity:

"Second, because the drivers of estuarine flooding are factorised through the multivariate distribution, it becomes easier to incorporate the effects of future changes. This is particularly the case if one is able to assume that the dependencies between variables are either not greatly affected by climate change or that changes in dependencies produce second-order effects on flood probability compared to changes in the marginal distributions. "

**Specific comment 13**

Line 272: The downscaling approach in Bevacqua et al. (2017) which related the water level in a 'joint probability zone' to the meteorological forcing's as a way of accounting for climate change may be of interest.

Response:

Thank you for this suggestion. This reference has been added.

**Specific comment 14**

Line 283: This sentence is very long and discusses two related but distinct issues. Please divide into two sentences.

Response:

This sentence has been broken into two sentences.

**Specific comment 15**

Line 284: Consider adding MacPherson et al. (2019) here as another method of accounting for the temporal shape of surge peaks in stochastic modelling and Environment Agency (2019) for an example where a single shape is derived to represent the largest surge peaks at a site.

Response:

The two references have been added.

**Specific comment 16**

Line 287: I suggest adding a reference to a review of the numerical models used to study compound flooding by Santiago-Collazo et al. (2019) here.

Response:

This reference has been added.

**Specific comment 17**

Line 296: Typo. Missing an "of" after "range".

Response:

Thank you for this observation. This has been corrected.

**Specific comment 18**

Lines 299: Grammar could be improved at the end of the sentence which starts on this line. Figure 3: Caption needs improving e.g. need to state what the colors of the points denote. Also, it is not clear why the Swan-Avon basin is split into two sections.

Response:

The sentence has been changed to:

"However, there are a few stream flow gauges near the outlet of the catchment but outside of the zone of tidal influence. These gauges include the Walyunga stream gauge and the Great Northern Highway stream gauge and are shown in Figure 3."

The caption is also revised as suggested.

**Specific comment 19**

Line 309: If URS is an acronym it needs to be defined.

Response:

It is a name, not an acronym.

**Specific comment 20**

Line 321: Also commonly referred as the 'transition zone' which could be added here.

Response:

This has been added as suggested.

**Specific comment 21**

Line 331: Poor grammar. The term "good quality" is not defined, and it should be made clearer the numbers at the end of the sentence refer to water levels. Is the data missing randomly throughout the series or is there a pattern e.g. missing values only occur during storms? This should be explored.

Response:

The data are missing or wrong when the gauge is out of order. It is not related to storms. The sentence has been revised:

"This leads to about 22 years of data with no missing or erroneous values, and with water levels ranging from 0.06 m to 1.92 m."

**Specific comment 22**

Line 369: Is the Mrl plot method the approach used to find the GPD threshold in the other approaches listed?

Response:

Yes. As mentioned at the end of section 4.2 no Method 2, "the same GPD-based frequency analysis described under Method 1 is used …".

**Specific comment 23**

Line 379: "One advantage of using the peak-over-threshold model for Approach 2 is that censoring can be used to improve the efficiency of full continuous simulation using a 2D hydrodynamic model, as only values above certain high thresholds are fully accounted for." I am a little confused here as the explanation in the introduction implies all of the water levels will be simulated when applying this approach. I appreciate the censoring is a good idea.

Response:

Thank you for this comment. Censored continuous simulation is used here because the GPD based frequency analysis is used and values below the threshold value does not need to be fully simulated. This also makes Method 2 feasible in terms of computational time.

**Specific comment 24**

Line 387: "By selecting all of the time periods when at least one of the boundary conditions is above the pre-determined threshold, this approach aims to simulate all water levels H above a specified high threshold value." Moderately high values of both boundary conditions could produce high water levels above a specified highwater level threshold, but these will not be accounted for in the suggested approach.

Response:

It depends on what is considered moderate, i.e. the threshold values used for both drivers and the final threshold value determined for water level H. This is also why a buffer time of 12 hours was used – partially to account for flood drivers with moderate values. Based on preliminary analysis conducted, all water levels values above the determinised threshold value are simulated.

This point has been added in the revised manuscript to improve clarity.

"The use of a time buffer accounts for the travelling time of water in the hydrodynamic model, and further ensures that the periods when flood level H are above the suitable GPD threshold value (e.g. generated by combination of moderate flood driver levels) will be fully simulated."

**Specific comment 25**
Line 412: "a random sample of simulation period (e.g. 1,000 hours)" Is this a continuous 1,000 hour period?

Response:
Not exactly. It includes a few different continuous periods that are in the low water level period.

**Specific comment 26**
Line 441: "In total, 28 flood events with flood drivers", why 28 events?

Response:
There are in total 15 historical events available. However, most of them cover case when one of the flood driver is extreme. In order to have a symmetric response surface, an additional 13 events were generated using scaled up historical flood driver data – they are like design events. The selection os events is discussed in section 4.3. A summary of these events is provided in the supporting material.

**Specific comment 27**
Line 459: "Finally, flood levels at the locations of interest (Step 3) are superimposed onto the bivariate dependence model 460 (Step 2) to estimate associated return periods." Not clear.

Response:
This section has been changed to:

"In the fourth and final step, the probability of different compound flood levels simulated in Step 3 can be derived based on the bivariate dependence model developed in Step 2 using the bivariate integration method introduced by Zheng (2015a). More details of this integration method can be found in Zheng et al. (2015b)."

**Specific comment 28**
Line 524: How are the independence and full dependence return periods calculated? Once added consider rearraiging some text so that Figure 9 is discussed in the same paragraph in which it is introduced.

Response:
The case of complete dependence/independence can be estimated by using different alpha values representing complete dependence (i.e. alpha = 0) and complete impendence (i.e. alpha = 1). This is discussed in section 4.3. Additional reference to section 4.3 has been added to improve clarity.

**Specific comment 29**

Line 531: Results reported in this paragraph are similar to those in Moftakhari et al. (2019) and Serafin et al. (2019) and probably elsewhere which could be cited here.

**Response:**

These references have been added.

**Specific comment 30**

Line 551-554: "This is very likely due to the systematic difference between the observed flood level data (with a maximum value of 1.92 m within the 22 years' data) and flood levels simulated using the MIKE21 model (with a maximum level of 1.86 m within the 31 years' analysis period) at this location." Interesting, is this due to a shortcoming of the MIKE21 model or the (short) distance between the two locations?

**Response:**

This could be a combination of both. The following comment has been added to improve clarity:

"In addition, the (short) distance between the tide gauge and the modelling location Sw10 could also be a contributing factor to this difference."

**Specific comment 31**

Line 572-574: "This over-estimation of flood levels for a given return period from Method 3 can potentially lead to over-conservative estimation of flood risk and costly flood prevention infrastructure." Or does using method 2 under-estimate flood levels and lead to under design?

**Response:**

There is no evidence that Method 2 underestimates flood levels. It is well known that Method 3 over-estimates flood levels, due to the assumption of static ocean levels and associated assumptions discussed in section 2.4. and section 6.

Additional discussion on this is added at the end of section 5.4:

"This over-estimation of flood levels for a given return period from Method 3 due to the use of a static tail water level and the associated assumption that the peaks of the two flood drivers with always concede can potentially lead to over-conservative estimation of flood risk and costly flood prevention infrastructure. "

**Specific comment 32**

Table 2: "Generally more difficult to account for a large number of flood drivers." Please expand on this. There are methods which allow the extension to more variables without restrictive assumptions regarding the nature of the dependence.

**Response:**

This is discussed in section 5.4. For example the current DVM can only account for two flood drivers.

**Specific comment 33**

Table 2: "Can be used to assess future conditions with dependence structure reflecting future changes". Very general and also could be true for approach 2 if the continuous simulation was run for future projected climatic conditions?

**Response:**

The key advantage of method 3 compared to method 2 is that time consuming hydrodynamic model runs can be avoided. This point has been added to improve clarity.

**Specific comment 34**

Line 583: Perhaps "established" is more suitable than "well-developed"?

**Response:**

This has been changed as suggested.

**Specific comment 35**

Line 607: "maintain key dependence between the boundary conditions". The dependence may change with time. I would highlight the fact that the method has the potential to account for climate change (unlike approach 1) as a benefit of the approach. The fact that the data is readily available is more of an (important) aside rather than a strength or limitation of the model.

**Response:**

This section has been changed to:

"…that reflect key dependence between the boundary conditions (e.g. of rainfall and the wind/pressure data that drive storm surge). Although, these high-resolution and temporally consistent data are at present not widely available under future climate scenarios, they can potentially be developed in the future allowing Approach 2 to be used to assess compound flood probability under future changes."

**Specific comment 36**

Line 611-613: "By separating the dependence estimation from the flood probability estimating process, future flood probability can be estimated by updating the dependence structure between flood drivers under these conditions without the requirement of additional flood simulation runs." Again, under the assumption that the dependence structure remains stationary.

**Response:**

This section points out that no additional hydrodynamic runs are required, while the dependence structure can be 'updated' to reflect future changes. Comments on the stationary assumption have been added in section 2.4.

**Specific comment 37**

L616-622: Some mention of this should go at the end of the results section.

**Response:**

Discussion on advantages/limitations of different methods are included section 6 not the results section. This is consistent for all three methods used. Therefore, this discussion over Method 3 is not included in the results section.

---

## Referee Report (RR1)

**Review of the revised paper: "Estimating the Probability of Compound Floods in Estuarine Regions Wu" by Wu et al.**

I thank the author for considering my comments. Overall, I found the paper improved and reccomend it for pubblicaiton. However, I have some minor additional technical comments.

Specific comment 37

L 509, Why do you use the values maximising the dependence? Understanding when the dependence is maximised provides interesting information on the physical system, however, the dependence values that are relevant from a point of view of the impact is that between the variables at the same time. In fact, the storm tide and the river flow interact at the same time in the real world.

Response:

This is because in Method 3 the information on the temporal dynamics (i.e. relative timing) of storm surges and astronomical tides is discarded and only the peaks of flood drivers are considered via the use of a static tail water level, as discussed in section 2.4. This is one of the limitations of Approach 3 and thus, Method 3. The following statement has been added to improve clarity:

"This is because in this method the information on the temporal dynamics of storm surges and astronomical tides is discarded and only the only the peaks of flood drivers and their joint dependence are considered, as discussed in section 2.4."

In practice and in general, could using the lag that maximise the depndence lead to importantly overestimate the risk? If so (as I believe), such a potential overestimation should at least be stressed explicitly (in addition to noticing the related limitation) in the text.

Specific comment 38

Fig 8, The 2D simulations receive as input time series of T and Q, therefore a question arises: which is the value of the time series that you consider as that to be reported on the x and y axes?

The plots, e.g., panel c, suggests that for a given 10 year return level of Q, when T becomes larger (from 0 to 1-year return period), H decreases. This is physically inconsistent. Such inconsistent behaviours seem to occur in the range of T AND Q below 1-year return levels. Do you have an explanation for that? If the explanation is convincing, one would then consider not showing values in this bivariate range (up to 1-year return level for both variables).

Response:

This variation is potentially caused by the interpolation method used. Additional discussion has been added in the revised manuscript to explain this.

"It can also be observed in Figure 8 that there are some variations in estimates of flood levels with very short return periods (e.g. return periods of 1 in 1 year or below), with the increase in one flood driver leading to decreased compound flood levels. Careful inspection of the results shows that this feature does not apply to any of the simulated data points, in the sense that simulation points with larger values of the boundary conditions always yield larger flood levels. Rather, the 'inflection' only occurs in a sparsely sampled region of the plot, and is thus suggestive of the limitations of using a log-linear interpolation scheme in this region. This therefore highlights the importance of carefully considering the sampling scheme as part of the analysis."

My suggestion to editor and authors is to add some hatching in the part of the plot that is not considered trustable such to highlight the issue to the reader. As all my comments, this is in the interest of the authors given that some readers may focus on the image at first and then on the text; hence, the image could look odd to the reader as unphysical. If that is computationally expensive, Ithink that the non trustable area should at least be highlighted in words also in the caption, where the author would refer to te text for further explanations.

The paper that I originally suggested (Bevacqua et al. (2020)) is now published at
https://www.nature.com/articles/s43247-020-00044-z

I believe that the work is relevant for some relevant statements that the author make about
the changes in the dependencies, i.e.

> *"This is particularly the case if one is able to assume that the dependencies between
> variables are either not greatly affected by climate change or that changes in
> dependencies produce second- order effects on flood probability compared to
> changes in the marginal distributions."*

This is the only work available in the literature where changes in both marginal and
dependencies of the meteorological drivers of compound flooding was considered for
Australia. Therefore it would serve as a basis some for the statements and I would suggest
considering it.

**Specific comment 24**

L 390, It is not clear to me why you need to account for the low water level periods through the
resampling approach, given that you will fit the GPD only to the extremes. I understand that is
necessary to be aware of the time in between the peaks to estimate the return periods, but why
simulating it?

Response:

One important reason that flood data during low water level periods are also 'simulated' using the
resampling approach is because the actual threshold values that will be used to fit the GPD is not
known a priori. The resampling approach will provide a reasonable transition of flood levels between
'flood periods' and 'low water level periods' compared to just using zero values and makes sure
reasonable flood level estimates will be used for flood probability estimation.

Thanks for the explanation, please also explain this in the paper if this is not done already.

**Specific comment 12**

L 185, During a discussion among colleagues, it was hypothesised that this may be related to the fact
that often there is interest in measuring either the sea level or the river discharge and therefore no
stations are collocated at the interface between the two. What do you think about this? Discuss it if
you think that this is relevant. I guess that this appears also discussed/hypothesised in Paprotny et
al. ("Compound flood potential in Europe").

Response:

Thank you for this suggestion, the following comment has been added.
"The lack of gauges within estuaries are likely to be at least in part due to the fact that there has
historically been greater interest in measuring either the sea level or the river discharge and
therefore there is less interest to place stations at the interface between the two (Paprotny et al.,
2018)."

Sorry for that, but I realize that the paper that I suggested to cite here (Paprotny et al.) was
not accepted for publicaiton, so I am not sure whether the journal allows for citing it. You
could refer to Bevacqua et al., 2017 (already cited in the paper) who also discuss the same
issue.

L368, Authors tend to oppose GPD and GEV as alternative approaches. Do you expect any differences in terms of uncertainties? Also, you use the GPD to estimate return periods/level. Shouldn't you also provide an equation for that?

Response:

The difference in the estimation outcomes from GPD vs GEV is out of the scope of this paper. The equation for the GPD is included in section 4.1.

I simply meant to add an equation for the return period based on the GPD (given that the GPD equation is provided). This seems not in the paper. Authors and editor can judge whether the reader would benefit from such an equaition or not.

Best regards.

L368, Authors tend to oppose GPD and GEV as alternative approaches. Do you expect any differences in terms of uncertainties? Also, you use the GPD to estimate return periods/level. Shouldn't you also provide an equation for that?

Response:

The difference in the estimation outcomes from GPD vs GEV is out of the scope of this paper. The equation for the GPD is included in section 4.1.

---

## Author Response (AR2)

**Response to reviewer 1 comments:**

**Comment 1:**

**Specific comment 37**

L 509, Why do you use the values maximising the dependence? Understanding when the dependence is maximised provides interesting information on the physical system, however, the dependence values that are relevant from a point of view of the impact is that between the variables at the same time. In fact, the storm tide and the river flow interact at the same time in the real world.

Response:

This is because in Method 3 the information on the temporal dynamics (i.e. relative timing) of storm surges and astronomical tides is discarded and only the peaks of flood drivers are considered via the use of a static tail water level, as discussed in section 2.4. This is one of the limitations of Approach 3 and thus, Method 3. The following statement has been added to improve clarity:

"This is because in this method the information on the temporal dynamics of storm surges and astronomical tides is discarded and only the only the peaks of flood drivers and their joint dependence are considered, as discussed in section 2.4."

In practice and in general, could using the lag that maximise the dependence lead to importantly overestimate the risk? If so (as I believe), such a potential overestimation should at least be stressed explicitly (in addition to noticing the related limitation) in the text.

**Response:**

This is a well-documented limitation of Approach 3, which is based on flood events. See (Zheng et al., 2015a; Zheng et al., 2015b; Zheng et al., 2013). As suggested by the reviewer, this limitation is mentioned at multiple locations in the manuscript, including section 2.4 on Approach 3[1] and Section 5.4 on results comparison[2] and Section 6 on discussion[3].
* * *
[1] Line 300: "Despite these advantages, there are several simplifications involved in this approach when converting continuous meteorological data into a set of multivariate 'design events', which could lead to significant misspecification of flood probability if not taken into account. … During this process information on the temporal dynamics of storm surges and astronomical tides is discarded. … a significant difficulty arises when trying to align the timing of the storm surge and astronomical tide events with the timing of the flood-producing rainfall in the upstream catchments Santiago-Collazo, F. L., Bilskie, M. V., and Hagen, S. C.: A comprehensive review of compound inundation models in low-gradient coastal watersheds, Environmental Modelling & Software, 119, 166-181, 2019.. Indeed, this problem has not been resolved, with most current methods using a stochastic method to account for the temporal shape of surge peaks MacPherson, L. R., Arns, A., Dangendorf, S., Vafeidis, A. T., and Jensen, J.: A Stochastic Extreme Sea Level Model for the German Baltic Sea Coast, Journal of Geophysical Research: Oceans, 124, 2054-2071, 2019. or taking a simplified approach such as assuming 'static' lower boundary conditions rather than explicitly resolving the tidal dynamics Zheng, F., Leonard, M., and Westra, S.: Application of the design variable method to estimate coastal flood risk, Journal of Flood Risk Management, doi: 10.1111/jfr3.12180, 2015a. 2015a.. The extent to which this simplification leads to mis-specified flood risk (and whether this misspecification leads to an under- or over-estimation of probabilities) is not known."

[2] Line 629: "... However, in the joint probability zone (e.g. locations Sw10 and Sw12) where both flood drivers have a significant impact on resulting flood levels, the event-based Method 3 results in significantly higher flood levels for a given return period compared to Method 2. This is especially the case for location Sw12, where flood levels estimated using Method 3 are above the upper bound of the 95% confidence interval generated using Method 2 based on censored continuous simulation data. This over-estimation of flood levels for a given return period from Method 3 due to the use of a static tail water level and the associated assumption that the peaks of the two flood drivers with always concede can potentially lead to over-conservative estimation of flood risk and costly flood prevention infrastructure."

[3] Line 678: "... However, by translating continuous flood time series data into a set of 'flood events', the information on coincident timing between different flood drivers is often lost, and various simplifying assumptions often need to be made. For example, when implementing the design variable method (DVM), the tail water level is assumed to be static (i.e. no tidal dynamics) with a value that corresponds to the specified

**Comment 2:**

**Specific comment 38**

Fig 8, The 2D simulations receive as input time series of T and Q, therefore a question arises: which is the value of the time series that you consider as that to be reported on the x and y axes?

The plots, e.g., panel c, suggests that for a given 10 year return level of Q, when T becomes larger (from 0 to 1-year return period), H decreases. This is physically inconsistent. Such inconsistent behaviours seem to occur in the range of T AND Q below 1-year return levels. Do you have an explanation for that? If the explanation is convincing, one would then consider not showing values in this bivariate range (up to 1-year return level for both variables).

**Response:**

This variation is potentially caused by the interpolation method used. Additional discussion has been added in the revised manuscript to explain this.

"It can also be observed in Figure 8 that there are some variations in estimates of flood levels with very short return periods (e.g. return periods of 1 in 1 year or below), with the increase in one flood driver leading to decreased compound flood levels. Careful inspection of the results shows that this feature does not apply to any of the simulated data points, in the sense that simulation points with larger values of the boundary conditions always yield larger flood levels. Rather, the 'inflection' only occurs in a sparsely sampled region of the plot, and is thus suggestive of the limitations of using a log-linear interpolation scheme in this region. This therefore highlights the importance of carefully considering the sampling scheme as part of the analysis."

My suggestion to editor and authors is to add some hatching in the part of the plot that is not considered trustable such to highlight the issue to the reader. As all my comments, this is in the interest of the authors given that some readers may focus on the image at first and then on the text; hence, the image could look odd to the reader as unphysical. If that is computationally expensive, I think that the non-trustable area should at least be highlighted in words also in the caption, where the author would refer to the text for further explanations.

The paper that I originally suggested (Bevacqua et al. (2020)) is now published at https://www.nature.com/articles/s43247-020-00044-z I believe that the work is relevant for some relevant statements that the author make about the changes in the dependencies, i.e. "This is particularly the case if one is able to assume that the dependencies between variables are either not greatly affected by climate change or that changes in dependencies produce second‐ order effects on flood probability compared to changes in the marginal distributions."

This is the only work available in the literature where changes in both marginal and dependencies of the meteorological drivers of compound flooding was considered for Australia. Therefore it would serve as a basis some for the statements and I would suggest considering it.

**Response:**

Thank you for the suggestion. The following wording has been added in the Caption:

"Note: the "inflection" in the contour lines for very short return periods is due to the use of interpolation scheme noting the sparsity of samples in these regions."

In addition, the recommended reference (Bevacqua et al., 2020) has been added in the revised manuscript.
* * *
exceedance probability. This simplifies the probability estimation process by assuming that the peak of tail water will always intercept with the peak of fluvial flood at any given location within the model domain, but it ignores the dynamic interactions of the two flood drivers, including the possibility that the peak fluvial flood wave will not occur at precisely the same time as the peak tidal cycle. Consequently, this method will always lead to over-estimation of flood levels Zheng, F., Leonard, M., and Westra, S.: Application of the design variable method to estimate coastal flood risk, Journal of Flood Risk Management, doi: 10.1111/jfr3.12180, 2015a. 2015a., as have been observed from results for the case study system."

**Comment 3:**

**Specific comment 24**

L 390, It is not clear to me why you need to account for the low water level periods through the resampling approach, given that you will fit the GPD only to the extremes. I understand that is necessary to be aware of the time in between the peaks to estimate the return periods, but why simulating it?

**Response:**

One important reason that flood data during low water level periods are also 'simulated' using the resampling approach is because the actual threshold values that will be used to fit the GPD is not known a priori. The resampling approach will provide a reasonable transition of flood levels between 'flood periods' and 'low water level periods' compared to just using zero values and makes sure reasonable flood level estimates will be used for flood probability estimation.

Thanks for the explanation, please also explain this in the paper if this is not done already.

**Response:**

Thank you for the suggestion. It has been added in the manuscript[4].

**Comment 4:**

**Specific comment 12**

L 185, During a discussion among colleagues, it was hypothesised that this may be related to the fact that often there is interest in measuring either the sea level or the river discharge and therefore no stations are collocated at the interface between the two. What do you think about this? Discuss it if you think that this is relevant. I guess that this appears also discussed/hypothesised in Paprotny et al. ("Compound flood potential in Europe").

**Response:**

Thank you for this suggestion, the following comment has been added.
"The lack of gauges within estuaries are likely to be at least in part due to the fact that there has historically been greater interest in measuring either the sea level or the river discharge and therefore there is less interest to place stations at the interface between the two (Paprotny et al., 2018)."

Sorry for that, but I realize that the paper that I suggested to cite here (Paprotny et al.) was not accepted for publication, so I am not sure whether the journal allows for citing it. You could refer to Bevacqua et al., 2017 (already cited in the paper) who also discuss the same issue.

**Response:**

Thank you for the suggestion. The reference has been updated.
* * *
[4] Line 442: "Since water level information below the selected threshold for fitting a GPD is censored in the frequency analysis, a resampling approach is used to fill in water level information during the low water level periods, which also addresses the challenging of not knowing *a priori* the exact value of the boundary condition thresholds."

**Comment 5:**

Specific comment 22

L368, Authors tend to oppose GPD and GEV as alternative approaches. Do you expect any differences in terms of uncertainties? Also, you use the GPD to estimate return periods/level. Shouldn't you also provide an equation for that?

Response:

The difference in the estimation outcomes from GPD vs GEV is out of the scope of this paper. The equation for the GPD is included in section 4.1.

I simply meant to add an equation for the return period based on the GPD (given that the GPD equation is provided). This seems not in the paper. Authors and editor can judge whether the reader would benefit from such an equation or not.

**Response:**

Thank you for the clarification. Estimation return period using a frequency analysis is a very common approach and there are many existing R/Python Libraries include functions that can do it. Personally, I like to reduce the number of equations in a paper, as too many equations can be a distraction. As a result, an additional equation on this is not added.

References:

MacPherson, L. R., Arns, A., Dangendorf, S., Vafeidis, A. T., and Jensen, J.: A Stochastic Extreme Sea Level Model for the German Baltic Sea Coast, Journal of Geophysical Research: Oceans, 124, 2054-2071, 2019.

Santiago-Collazo, F. L., Bilskie, M. V., and Hagen, S. C.: A comprehensive review of compound inundation models in low-gradient coastal watersheds, Environmental Modelling & Software, 119, 166-181, 2019.

Zheng, F., Leonard, M., and Westra, S.: Application of the design variable method to estimate coastal flood risk, Journal of Flood Risk Management, doi: 10.1111/jfr3.12180, 2015a. 2015a.

Zheng, F., Leonard, M., and Westra, S.: Efficient joint probability analysis of flood risk, Journal of Hydroinformatics, 17, 584-597, 2015b.

Zheng, F., Westra, S., and Sisson, S. A.: Quantifying the dependence between extreme rainfall and storm surge in the coastal zone, Journal of Hydrology, 505, 172-187, 2013.

**Response to reviewer 2 comments:**

**Comment 1:**

L64-L66: Slightly unclear. "experiencing long term changes" Are these the long-term changes caused by the aforementioned long-term climate phenomena. If not, please provide more details of their sources.

**Response:**

The statement here intended to refer to the fact that "the joint probability of flood drivers" are changing over time, which has been reported in the two reference provided, i.e. Arns et al., 2020 and Bevacqua et al., 2019. To our knowledge these changes have not yet been formally attributed either to climate change or other processes, so it is not possible to provide a causal statement.

**Comment 2:**

L80: "Superposition on the astronomic tide". The interaction of the surge and tide could be mentioned explicitly here.

**Response:**

Thank you for the suggestion. "i.e. the interaction of surge and tide," has been added in the sentence.

**Comment 3:**

Figure 2 and following paragraphs text: The text describing the approach could be simplified (in parts) by referring to the pathways outlined in Figure. It seems strange that the pathways discussed in the Figure are not mentioned at all in the text.

**Response:**

The details of the pathways can be in the caption or the text or both. After careful consideration, we decided to keep the description of the pathways in the caption, as there will be readers who may look at the figure without reading the text in detail. The reason we did not repeat the details of the pathways in the text is to avoid duplication. In addition, readers' understanding of the main text will not be compromised without these additional details in the text.

**Comment 4:**

L296: "two components mentioned above". The two components being referred to are not immediately obvious and should be stated. I assume you are referring to the probability/statistical modeling and hydrodynamic modeling.

**Response:**

Thank you for this observation. The sentence has been changed to "the two components indicated above (i.e. flood surface and associated probability)".

**Comment 5:**

Figure 3: Avon basin in the caption but Swan-Avon basin on the Map. Be consistent!

**Response:**

Thank you for this observation. It has been changed to "Swan-Avon basin".

**Comment 6:**

L412-413: "as only values above certain high thresholds are full accounted for". Is it not periods where there is at least one value about the high thresholds rather than only values above a certain threshold? Also "fully accounted for" is very vague please be more specific.

**Response:**

Thank you for this observation. It has been changed to "as only values above certain high thresholds need to be included as part of the joint probability calculation".

**Comment 7:**

L423: I think "both" rather than "either" would be more accurate.

**Response:**

There are situations where floods can be caused by one of the two drivers. Therefore, it should be "either".

**Comment 8:**

Figure 5: Please provide a more detailed explanation of how the 'high water level periods' are defined. For instance, is there always a gap between the end of the initial buffer period and the first exceedance of T during an event, why? How long is the gap? Does the post event buffer always begin once Q falls below its threshold?

**Response:**

The definition of "high water level period" is provided in the second paragraph of section 4.2 just above Figure 5:

*"The combination of the flood periods and the time buffer periods is referred to as the high water level periods,"*

The reason a buffer period is used is because the threshold values for both flood drivers are not known a priori, as mentioned at the beginning of the second paragraph in section 4.2. By extending the simulation period of floods, we increase the likelihood that all water levels above the required threshold values (which will be estimated after all simulations are complete) are simulated.

**Comment 9:**

L607: Grammar. Remove "relatively".

**Response:**

Thank you for the observation. It has been removed.

**Comment 10:**

L672: Dependence estimation does not include marginal distribution estimation. I would therefore change dependence estimation to "joint probability estimation" or similar.

**Response:**

"Dependence" is the terminology used in most relevant studies on the third approach, which is the focus of discussion at this location. Therefore, "dependence estimation" is used to be consistent with previous studies.

**Comment 11:**

L687: "2) the dependence between the two flood drivers is location specific" I am not sure how this is more of a disadvantage with this method compared with the other two approaches.

**Response:**

Out of all three approaches, only in the third approach the dependence is considered separately. Since the dependence is location specific and requires to be estimated using an appropriate statistical model, it is included here as a challenge for Approach 3 for the completeness of the statement. The statement has been revised as below to make it clearer: "… and 2) the dependence between the two flood drivers is location specific and needs to be estimated using an appropriate statistical model (Zheng et al., 2015a)"